# Comprehensive Evaluation and Selection of 192 Maize Accessions from Different Sources

**DOI:** 10.3390/plants13101397

**Published:** 2024-05-17

**Authors:** Mengting Hu, Huijuan Tian, Kaizhi Yang, Shuqi Ding, Ying Hao, Ruohang Xu, Fulai Zhang, Hong Liu, Dan Zhang

**Affiliations:** 1College of Agriculture, Tarim University, Alar 843300, China; humtww@163.com (M.H.); wmzthj@163.com (H.T.); yangkaizhi2022@163.com (K.Y.); wzcda150314@163.com (S.D.); HY18386110782@163.com (Y.H.); 17861086520@163.com (R.X.); zfl7127@163.com (F.Z.); 18893726646@163.com (H.L.); 2Key Laboratory of Genetic Improvement and Efficient Production for Specialty Crops in Arid Southern Xinjiang of Xinjiang Corps, Alar 843300, China

**Keywords:** maize, accessions, genetic diversity, comprehensive evaluation

## Abstract

In the period 2022–2023, an analysis of fourteen phenotypic traits was conducted across 192 maize accessions in the Aral region of Xinjiang. The Shannon–Wiener diversity index was employed to quantify the phenotypic diversity among the accessions. Subsequently, a comprehensive evaluation of the index was performed utilizing correlation analysis, principal component analysis (PCA) and cluster analysis. The results highlighted significant findings: (1) A pronounced diversity was evident across the 192 maize accessions, accompanied by complex interrelationships among the traits. (2) The 14 phenotypic traits were transformed into 3 independent indicators through principal component analysis: spike factor, leaf width factor, and number of spikes per plant. (3) The 192 materials were divided into three groups using cluster analysis. The phenotypes in Group III exhibited the best performance, followed by those in Group I, and finally Group II. The selection of the three groups can vary depending on the breeding objectives. This study analysed the diversity of phenotypic traits in maize germplasm resources. Maize germplasm was categorised based on similar phenotypes. These findings provide theoretical insights for the study of maize accessions under analogous climatic conditions in Alar City, which lay the groundwork for the efficient utilization of existing germplasm as well as the development and selection of new varieties.

## 1. Introduction

Maize (*Zea mays* L.) occupies a pivotal role in the agricultural history of China, contributing significantly to animal husbandry, light industry, energy, and related sectors [1]. A series of unique accessions have gradually accumulated with the continuous promotion of maize cultivation in China [2,3]. These resources contain rich genetic diversity and offer great potential for maize breeding [4,5]. The varied climatic conditions in Xinjiang, particularly in its southern region, along with significant ecological variability across different locales, necessitate high adaptability and stability from maize accessions. Investigating the growth attributes of maize accessions and identifying superior germplasm are essential for providing theoretical foundations to tackle production challenges across diverse ecological zones. The variability in phenotypic traits, encompassing agronomic characteristics, stems from the interplay between genetic diversity and environmental factors such as climate, soil type, fertility, and cultivation practices across different years and locations [6,7,8]. Phenotypic traits, reflecting gene–environment interactions, are readily available and so convenient for evaluation [9]. Enhancements in plant structure can notably augment maize kernel yield and, as noted by Malik et al. and Rahman et al., critical agronomic traits for maize include plant height, ear height, stem thickness, leaf width, leaf length, total leaf count, ear count per plant, thousand kernel weight, and male ear branching [10,11]. Optimal plant height and suitable ear height significantly enhance resistance to lodging. The angle between stem and leaf, along with leaf dimensions, is integral to the spatial arrangement of maize and its capacity for light capture [12,13,14,15]. Research indicates that varying stem and leaf angles impact maize’s density tolerance and overall yield differently. [16,17]. In addition, a balanced leaf display maximizes light utilization, thereby boosting photosynthesis and significantly fostering higher maize yields [18]. The structure of the male ear, particularly the branch count, also plays a crucial role in yield outcomes [19]. Maize plants with male ear branches removed had higher yields than intact plants [20]. Hunter demonstrated that at elevated planting densities, smaller male spikes enhance productivity by reducing light obstruction and allocating more energy towards grain production [21]. Thus, understanding trait interrelations is vital for effective selection in breeding programs, aiding breeders in recognizing and exploiting diverse germplasm resources.

Numerous studies have focused on the morphological identification and evaluation of maize accessions. Li et al. established the core germplasm of local maize varieties in China, selecting based on geographic origin and phenotypic diversity. Their findings highlighted that varieties from the mountainous southwest exhibited significant phenotypic diversity [22,23]. Cai Yi-Lin analyzed the phenotypic diversity of 710 maize accessions by geographic origin, revealing notable regional differences in diversity indices and traits, with germplasm from South China, East China, and Southwest China showing higher diversity than other regions [24]. Dong Xin et al. conducted an extensive phenotypic assessment of 129 local maize varieties in Chongqing province over two years, organizing them into three main groups [25]. Meng Zuqing et al. evaluated the phenotypic diversity of 179 local maize varieties in Tibet through multi-year trials, laying the groundwork for the preservation and development of these varieties in Tibet [26]. Analyzing accessions is essential for advancing research that seeks to elucidate their genetic backgrounds. Larik et al. demonstrated that any breeding program must begin with an evaluation of germplasm, a vital source of genetic variability [27]. Assessing the extent of phenotypic genetic diversity in maize germplasm resources can minimize redundancy in conservation efforts, aid in the development of core germplasm collections, and enhance the effective utilization of genetic resources in breeding programs [28].

There have been few studies on screening maize accessions in Xinjiang within the field of domestic maize accessions. This study focused on evaluating 14 traits of 192 maize accessions introduced to the Aral region of Xinjiang during the years 2022–2023. The objective was to investigate the growth habits and characteristics of its features in the Aral region of Xinjiang.

## 2. Results

### 2.1. Distribution Pattern of Phenotypic Traits of Maize Accessions from Different Sources

To examine the distribution patterns of various traits across maize accessions over two years, both bar charts and curve fitting were employed to illustrate the trait distributions, facilitating visualization of their distribution patterns. As depicted in Figure 1, the frequency distribution for each agronomic trait typically presents a central peak with lower frequencies at both ends. Notably, the frequency distribution ranges for traits such as spike leaf width, leaf width at the upper ear, number of leaves, and number of tassel branches exhibited minimal variation between the two years. Conversely, traits like ear height, chlorophyll content, and the number of effective spikes demonstrated more significant variations over the two years. Moreover, the frequency distribution ranges for most remaining traits were higher in 2022 compared to 2023, indicating that environmental factors exerted a relatively minor influence on the distribution of traits such as spike position, spike leaf width, leaf width at the upper ear, number of leaves, and number of tassel branches. Nonetheless, for the majority of traits, environmental conditions did play a considerable role in influencing their frequency distributions.

### 2.2. Diversity Analysis of Phenotypic Traits in Maize Accessions

The diversity index can more accurately reflect the genetic diversity of resources, with a higher index indicating greater diversity among traits [29]. An analysis of the agronomic trait data of 192 maize germplasms over two years revealed that, aside from plant height and stem–leaf angle, which exhibited significant mean value differences between the years, the mean values of other traits did not differ significantly(Table 1). In 2022, all traits, with the exception of ear height and stem–leaf angle, registered diversity indices above 2.000, with leaf width at the upper ear achieving the highest diversity index (H’ = 2.093), closely followed by spike leaf length (H’ = 2.092). In 2023, all traits, apart from spike leaf length, had diversity indices above 2.000, with the tassel branch number displaying the highest diversity (H’ = 2.086), and stem diameter next (H’ = 2.081). The coefficient of variation (CV) is indicative of the extent of variability for each trait among maize accessions [30], with a CV greater than 10% suggesting substantial differences between varieties [31]. In this study, the CV for all traits, except for the effective ear tip, exceeded 10% in both years. Ear height presented the highest CV at 38.20% and 32.27%, respectively, followed by tassel branch number at 32.11% and 30.37%. Additionally, the CVs for ear height/plant height ratio, upper ear leaf width, number of leaves, and tassel weight showed consistency between the two years, although a significant disparity in ear height across the years was observed. In summary, the analysis of 192 maize accessions highlighted considerable phenotypic trait differences in the field, revealing a rich pool of variation within the accessions. Notably, the coefficients of variation for ear height (38.20%, 32.27%) and tassel branch number (32.11%, 30.37%) were above 30%, indicating a high level of variability and substantial potential for genetic enhancement.

### 2.3. Analysis of Variance (ANOVA) for Phenotypic Traits of Maize Accessions

An ANOVA analysis of two-year data on 14 phenotypic traits of maize accessions revealed significant differences between years and among different maize accessions for all traits, with the exception of ear height and thousand kernel weight (Table 2). These findings suggest notable disparities in environmental conditions between the two years, as well as significant genetic trait differences among the varieties. The statistical analysis indicated significant interactions between years and varieties for all traits, highlighting the extensive influence of environmental conditions on the varieties.

### 2.4. Correlation Analysis of Maize Accessions for Various Phenotypic Traits

A correlation analysis of 192 maize accessions, based on 14 phenotypic traits, was performed to assess the strength of association among these traits (referenced in Figure 2). Highly significant positive correlations were observed among plant height, ear height, the ratio of ear height to plant height, spike leaf length, spike leaf width, length and width of the upper ear leaf, and leaf number. With the exception of leaf number, tassel branch number, stalk diameter, thousand kernel weight, and chlorophyll content, effective spikes demonstrated significant to highly significant positive correlations with the other traits. Thousand kernel weight was positively correlated with all traits except for effective spikes, tassel branch number, and stem–leaf angle. There was no significant correlation between stem–leaf angle and tassel branch number; however, stem–leaf angle exhibited a significant to highly significant negative correlation with the other traits. Chlorophyll content was positively correlated with plant height, ear height, spike leaf length, spike leaf width, length and width of the upper ear leaf, leaf number, tassel branch number, and thousand kernel weight. Moreover, chlorophyll content showed a significant positive correlation with effective spikes.

### 2.5. Principal Component Analysis of Maize Accessions

The assessment of maize accessions may be influenced by the intricate interrelationships among traits. Principal component analysis (PCA) offers a more precise and standardized method for evaluating maize accessions by transforming multiple trait indicators into fewer composite indicators through dimensionality reduction [32,33]. The Kaiser–Meyer–Olkin (KMO) and Bartlett tests (KMO = 0.805 > 0.65, *p* < 0.01) demonstrated a high degree of correlation among factors, justifying the application of PCA to the data [34]. The first three principal components, each with an eigenvalue greater than 1.0, accounted for 31.53%, 24.29%, and 9.75% of the total variance, respectively. Their cumulative contribution rate of 65.57% suggests that these components encapsulate the majority of the genetic information for most traits in maize germplasm (Table 3). The first principal component had an eigenvalue of 4.41, with ear height and the ratio of ear height to plant height having the highest loadings. These variables are predominantly associated with ear positioning, thus referred to as the ear position factor. The second principal component, with an eigenvalue of 3.40, had its highest loadings on spike leaf width and the width of the upper ear leaf, primarily related to leaf width and consequently termed the leaf width factor. The third principal component had an eigenvalue of 1.37, with the effective spike exhibiting the highest and positive eigenvector, while the stem–leaf angle had the lowest and negative eigenvector, indicating a negative correlation between stem–leaf angle and effective spike.

### 2.6. Cluster Analysis of Maize Accessions

Cluster analysis can categorise genetically similar germplasm into a single group, thereby elucidating taxonomic characteristics and relationships [35]. At a Euclidean distance of 17 (as illustrated in Figure 3), 192 maize accessions can be categorized into three principal groups, as detailed in Table 4 and Table 5. Group I consists of 104 accessions, representing 54.17% of the total. This group includes 82 domestic and 22 international resources, ranking second in phenotypic trait performance relative to the other groups. The coefficients of variation for all traits exceed 10%, with the exception of effective spikes. Group II encompasses 32 accessions, making up 16.67% of the total, with 31 being domestic and one international. This group is characterized by the largest stem–leaf angle, and it exhibits the lowest values for the remaining traits among the three groups. All traits, aside from plant height and the effective number of spikes per plant, have coefficients of variation above 10%. Group III includes 56 accessions, accounting for 29.17% of the total, including 46 domestic and 10 international entries. This group demonstrates the best overall performance, with the lowest coefficient of variation among the three groups. Coefficients of variation for traits such as leaf width at the spike position, upper leaf width, effective number of spikes per plant, and stem thickness all fall below 10%. Additionally, this group has the lowest mean value for the stem–leaf angle. The origin of each group shows no significant difference, suggesting that the clustering of varieties is minimally related to their geographic origins.

## 3. Discussion

### 3.1. Diversity of Phenotypic Traits in Maize Promotes Selection of Accessions

Analysing the genetic diversity of phenotypic traits is fundamental to crop breeding work. Phenotypic traits are the most intuitive characteristics of plants [7]. Investigating the diversity of agronomic traits in maize accessions can uncover genetic variances among traits from diverse sources, facilitating the identification of germplasm with superior performance. This research compared the distribution patterns of 14 phenotypic traits across 192 maize resources over two years, analyzing the variation. The analysis revealed some variation in the distribution characteristics of the 14 traits across the years, attributed primarily to environmental factors. Notably, May 2023 experienced five days of low temperatures, and the annual precipitation decreased significantly compared to 2022. The differences also stemmed from the varied geographic origins of the maize germplasm. Except for plant height and stem–leaf angle, the mean values of other traits showed minimal differences between the years, indicating stable identification in this environmental context. The diversity analysis results demonstrated that the test material exhibits higher levels of phenotypic diversity, aligning with the findings of Ma et al. [36] and Meng et al. [26], but diverging from those of Sirlene Viana d.F [37] and Syahruddin K [38]. These discrepancies are likely due to variations in environmental conditions, climate, cultivation practices, and germplasm resource origins. Xinjiang’s southern border possesses unique geographical and climatic features, creating a complex and diverse ecological environment. This diversity in the maize growth process necessitates the introduction of accessions with good adaptability, laying the groundwork for future variety selection [39]. Plant architecture is intricately linked to canopy structure, lodging susceptibility, photosynthetic efficiency, and yield in maize [40,41,42]. In breeding practices, maintaining reasonable plant height and lower ear position is crucial for ensuring stable and high maize yields. The study identified highly significant positive correlations between plant height, ear height, and the ear-to-plant height ratio, corroborating previous research by Shen et al. [43] and Zhu et al. [44]. The morphology and distribution of maize leaves directly affect light interception by the canopy and the efficiency of photosynthetic utilization, thereby influencing yield formation [45]. Effective spikes and thousand kernel weight are key components of yield composition [46,47]. This study identified significant to highly significant positive correlations between the effective number of spikes, thousand kernel weight, and traits such as plant height, ear height, ear-to-plant height ratio, leaf length, leaf width, and chlorophyll content. The 192 maize germplasm resources analyzed exhibited substantial phenotypic diversity. Traits both interact and constrain one another; thus, enhancing one trait should be balanced with the careful selection of others. For instance, plant height, ear height, and stem thickness were found to be significantly positively correlated in this study. While selecting for resistance to lodging, particular attention must be paid to stem thickness; however, excessive plant and ear heights can also contribute to lodging, necessitating balanced control of these traits [48]. Therefore, these traits can be strategically selected to meet diverse breeding objectives.

### 3.2. Comprehensive Evaluation Is Conducive to the Selection of Superior Germplasm

Principal component analysis (PCA) offers high credibility and objective evaluation and is widely used in the evaluation analysis and comprehensive assessment of germplasm resources across various crops [49,50,51]. Furthermore, traits display complex interrelationships and constraints, leading to intricate relationships among them. Individual indicators alone cannot provide a complete and accurate evaluation; however, PCA addresses the issue of missing data by effectively reducing multiple variables to key factors [52]. Consequently, employing multivariate statistical methods to assess and screen composite indicators becomes particularly crucial. In this study, 14 phenotypic traits were condensed into three principal components: ear position factor, leaf width factor, and effective spike, achieving a cumulative contribution rate of 65.57%. This rate suggests that these components represent 65.57% of the variability in the 14 phenotypic traits across the 192 maize accessions. Consequently, this study selected ear height, ear position coefficient, leaf width at ear level, leaf width of upper leaves near the ear, and effective number of ears per plant as the core indices for evaluating maize phenotypic traits.

### 3.3. Selection of Accessions Lays the Foundation for Selection

In this study, 192 samples were categorized into three primary clusters through cluster analysis. Group III exhibited superior performance, ranking first among the three groups, making it the most favourable option for cultivation in the Aral region of Xinjiang and areas with similar climates. The phenotypes were characterized by taller plant height, broader leaves, thicker stalks, higher chlorophyll content, more effective spikes, increased thousand grain weight, and yield advantages. Leaves, stalks, and ears are crucial components contributing to the biological yield of silage maize [53]. Silage maize typically features tall, compact plants with wide leaves and substantial biomass [54]. Thus, it serves as a fundamental resource for selecting fodder varieties and as an excellent progenitor for material innovation and cross-breeding [55,56]. Group II, which ranked last in phenotypic trait performance, exhibited the largest coefficients of variation across all traits. It displayed the shortest plant and spike heights and the widest stem–leaf angle among the groups. This group could be beneficial in combination with materials characterized by excessive plant height, spike height, and narrow stem–leaf angles, to achieve an optimal plant type [57,58]. Such adjustments could lay the groundwork for plant type improvement and, due to its low spike position, foster the development of varieties with enhanced lodging resistance in conjunction with better-performing materials [56]. Group I, positioned second in overall performance, is also well suited for cultivation in the Aral region of Xinjiang and similar climates, offering substantial potential for agricultural exploitation. The uniformity in trait performance renders these samples solid foundational materials for variety selection [59,60]. The significant genetic variations observed among different taxa can enrich a variety of improvement processes. Consequently, the outcomes of this grouping can provide a reference and selection basis to meet diverse breeding objectives.

The analysis of 192 maize accessions demonstrated that varieties sharing the same designation originated from diverse sources. Through examining their genetic similarities, we identified 14 groups of materials with identical names and close genetic links, including Huang Corn (Nos. 153, 168), Bai Bao Gu (Nos. 86, 108), Bai Ma Ya (Nos. 73, 120), Bai Corn (Nos. 67, 129, 134, 136, 151), Da Qing Ke (Nos. 33, 41), Hong Gu Zi (Nos. 31, 40), Hong Corn (Nos. 74, 83, 130), and Huang Corn (Nos. 77, 107, 128, 133). The cluster analysis grouped materials with the same name and homologous origins, highlighting examples such as Nos. 77, 107, 128, and 133, Jin Ding Zi (Nos. 24, 29), Late Maturing Maize (Nos. 138, 149), Xiao Li Hong (Nos. 28, 58), Xiao Qing Ke (Nos. 36, 38, 39), You Zi Bai (Nos. 121, 125), and Corn (Nos. 51, 53). These materials are suspected to belong to the same variety. The occurrence of homologous or heterologous phenomena among these groups will be further elucidated using molecular techniques, contributing to the standardization and management of the collected accessions.

## 4. Materials and Methods

### 4.1. Materials and Sources

The maize accessions utilized in this study were sourced from the Institute of Crop Science, the Chinese Academy of Agricultural Sciences, and the National Crop Accessions Sharing Service Platform, comprising a total of 192 samples. Detailed information about these accessions is provided in Appendix A, with the sources of the various accessions illustrated in Figure 4.

### 4.2. Experimental Design

The experiment was conducted at the practical teaching base of Tarim University located in Alar City, Xinjiang (refer to Figure 4 for the exact location). Seeds were sown in fully moist soil on 21 April 2022, and 16 April 2023. Each accession was planted in three rows, maintaining a plant spacing of 30 cm and a row spacing of 60 cm. Mechanical mulching with perforation combined with manual spot sowing was employed. Each row contained ten holes, with each hole being thinned to retain one seedling. Each plot comprised three rows, covering an area of 5.4 m^2^. The experiment was replicated three times within a protected area to ensure consistency.

### 4.3. Research Area Climate Characteristics

South Xinjiang is located in the southern region of the Xinjiang Tianshan Mountain range, characterized by a typical continental arid climate with annual precipitation ranging from 20 to 100 mm. The average temperature in the South Xinjiang Plain varies between 10 °C and 13 °C, enjoying a frost-free period of 200 to 220 days. The area experiences significant diurnal temperature variations and low rainfall due to its arid conditions. Alar City, situated in the southern part of the Xinjiang Uygur Autonomous Region of China, lies on the northern edge of the Taklamakan Desert and the upper reaches of the Tarim River (80°30′~81°58′ E; 40°22′~40°57′ N). The city’s climate is classified as warm-temperate extreme continental arid desert. The average annual effective accumulated temperature at or above 10 °C is 4541.4 °C, with average annual sunshine hours ranging from 2556.3 to 2991.8 h [61]. Alar City benefits from ample light and heat, although it is occasionally subjected to dust storms.

#### 4.3.1. Irrigation and Fertiliser Application in the Trial Area

The soil at the experimental site is characterized by sandy loam texture. Irrigation was conducted using a tube with two rows of under-membrane drip tubes, aligned with the water demand law for corn [62]. The preceding crop at this location was cotton. Prior to sowing, a compound fertilizer at a rate of 525 kg/ha was applied, followed by an application of 270 kg/ha of urea one month post seedling emergence. Harvest occurred in early October of the same year upon reaching maturity. Practices concerning fertilizer application, cultivation management, and pest control in the experimental field mirrored those used in standard field conditions.

#### 4.3.2. Comparison of April–October Climate in the Test Area for Two Years

The peak temperatures in 2022 and 2023 were observed in July, reaching 34 °C and 35 °C, respectively. In 2022, the rainfall was primarily concentrated in July and August, amounting to a total of 32.2 mm. Conversely, in 2023, the rainfall was evenly spread from April to September, totalling 24.0 mm, which is 8.2 mm less than the previous year (Table 6).

In May 2023 (Figure 5), an extreme cold weather event occurred with temperatures dropping to as low as 0 °C on 5 May and remaining below 10 °C for five consecutive days. The average temperature during this period ranged between 3.5 °C and 16.5 °C. Data from Weather Forecast (https://tianqi24.com/alaer/history.html. Accessed on 5 November 2023).

### 4.4. Project Measurement

In this study, the phenotypic characteristics of 192 maize accessions were assessed in field conditions as per the Specification and Data Standard for Maize Accessions Description [63]. At maturity, key traits for these accessions were evaluated, including plant height (cm) [64], ear height (cm) [65], ear height to plant height ratio (%) [65], spike leaf length (cm) [66], spike leaf width (cm) [66], leaf length of upper ear (cm) [67], leaf width of upper ear (cm) [67], leaf number [68], effective spike [69], tassel branch number [70], stalk diameter (mm) [71], stem–leaf angle [72], thousand kernel weight (g) [73] and chlorophyll content [74]. A total of 14 traits were characterised.

Plant Height: Measurement from the ground to the tip of the male spike.

Ear Height: Distance from the ground to the first fruit set’s node.

Spike Coefficient: The ratio of ear height to plant height.

Spike Leaf Dimensions: Length and width of the spike leaf are measured from the base to the tip and across the widest part perpendicular to the midvein, respectively.

Upper Ear Leaf Dimensions: Similar measurements are taken for the leaf length and width of the upper ear.

Leaf Number: Count of total leaves from seedling emergence to post-male extraction.

Effective Spike: Calculation of effective ears per plant at harvest by counting total maize plants and ears.

Tassel Branch Number: Count of main and branching axes of the male spike.

Stalk Diameter: Measured at the middle of the third internode with vernier calipers.

Stem–Leaf Angle: Angle between the spike leaf and stalk measured with a protractor.

Thousand Kernel Weight: Weight of 1000 randomly selected mature kernels.

Chlorophyll Content: Assessed using a hand-held chlorophyll content meter (Spad-502Plus).

### 4.5. Data Analysis

#### 4.5.1. Shannon–Weaver Diversity Index

The observations were tropified into 10 groups based on the mean observations for each trait ($\bar{X}$) and standard deviation (σ), according to $\bar{X}$ ± kσ (k = 0, 0.5, 1, 1.5, 2) (Table 7). The graded data were then used to calculate the Shannon–Weaver diversity index using the formula H’ = −$\sum_{i=1}^{n}Pi\times In(Pi)$. The variable Pi represents the frequency of distribution of the i-th rank of a trait. This is calculated by dividing the number of materials in the i-th rank by the total number of materials [75].

#### 4.5.2. Trait Analysis

##### Descriptive Statistics

Descriptive statistics, including maximum, minimum, mean, standard deviation, coefficient of variation, and Shannon–Weaver diversity index for 14 phenotypic traits of 192 maize accessions, were calculated for two years (2022–2023) using Microsoft Excel 2021 software [76,77].

##### Frequency Distribution

Frequency distributions for each trait over the two years were generated using Origin 2021 [78]. Histograms were constructed with the centre of intervals and frequencies, and Gaussian fits in nonlinear curve fitting were applied to assess the variation in each trait across the two years [79].

##### Correlation Analysis

Data from both years were averaged, and correlation analysis was conducted using Origin 2021 software with the Correlation Plot plug-in. Heatmaps were generated at the 0.01 and 0.05 levels (Pearson method) [80].

##### Analysis of Variance (ANOVA)

Multivariate ANOVA was performed with IBM SPSS Statistics 27.0.1 to analyze relationships between materials, between years, and between material–year interactions [81].

##### Principal Component Analysis (PCA)

Principal component analysis on the averaged data from the two years was conducted using IBM SPSS Statistics 27.0.1 software, employing the Varimax method [82]. The data were standardized using the affiliation function, and the formula was applied to calculate the composite evaluation F-value [83].

##### Cluster Analysis

Systematic cluster analysis was executed with IBM SPSS Statistics 27.0.1 software, employing within-group linkage and using Euclidean distances to determine genetic distances [33].

##### Accession and Test Site Markers

Maps of accession and trial site markers were generated using ArcMAP 10.8 software, with site data sourced from Map Quest (https://map.bmcx.com/meiguo__map/). Accessed on 21 March 2024.

## 5. Conclusions

The 192 maize accessions exhibited high levels of diversity and variation. Over two years, the performance of each trait showed varying degrees of change. Correlation analysis revealed some correlations among the traits. Principal component analysis converts the 14 phenotypic traits to three core indicators. Through cluster analysis, the varieties were categorized into three groups. Based on the actual conditions, accessions beneficial to each group can be chosen as foundational materials for future breeding efforts.

## Figures and Tables

**Figure 1 plants-13-01397-f001:**
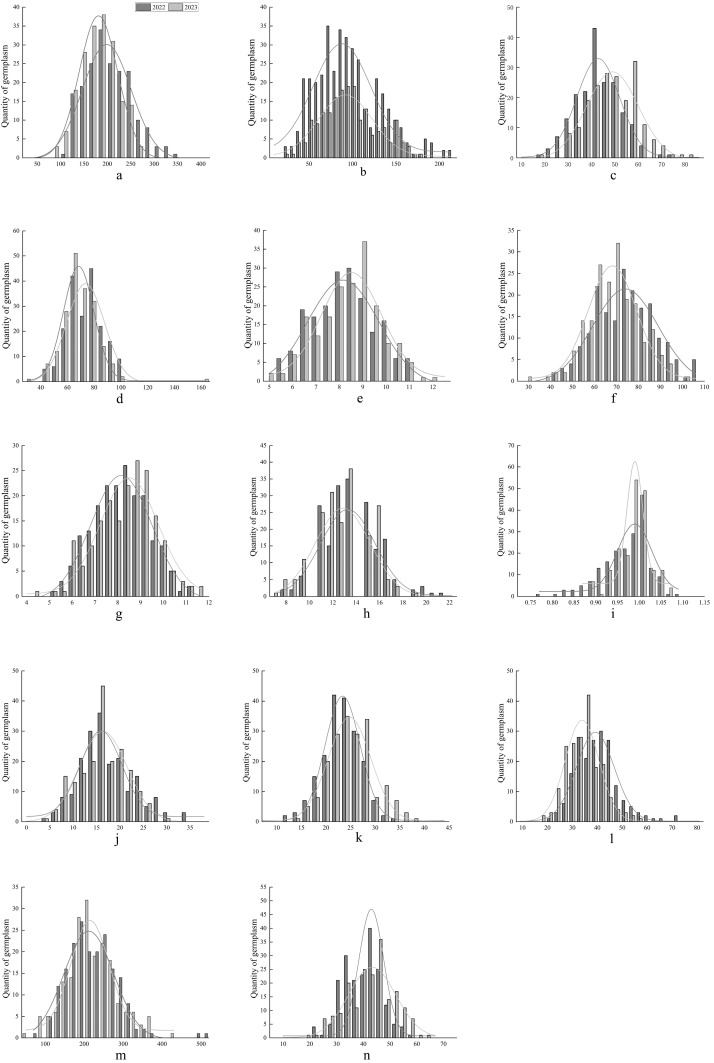
Distribution pattern of 14 phenotypic traits between the two years. Note: In the figure, (**a**–**n**) represent plant height, ear height, ear height to plant height ratio, spike leaf length, spike leaf width, leaf length of upper ear, leaf width of upper ear, leaf number, effective spike, tassel branch number, stalk diameter, stem–leaf angle, thousand kernel weight, and chlorophyll content. The trait value is represented on the horizontal axis, and the amount of germplasm in the interval for the trait is represented on the vertical axis.

**Figure 2 plants-13-01397-f002:**
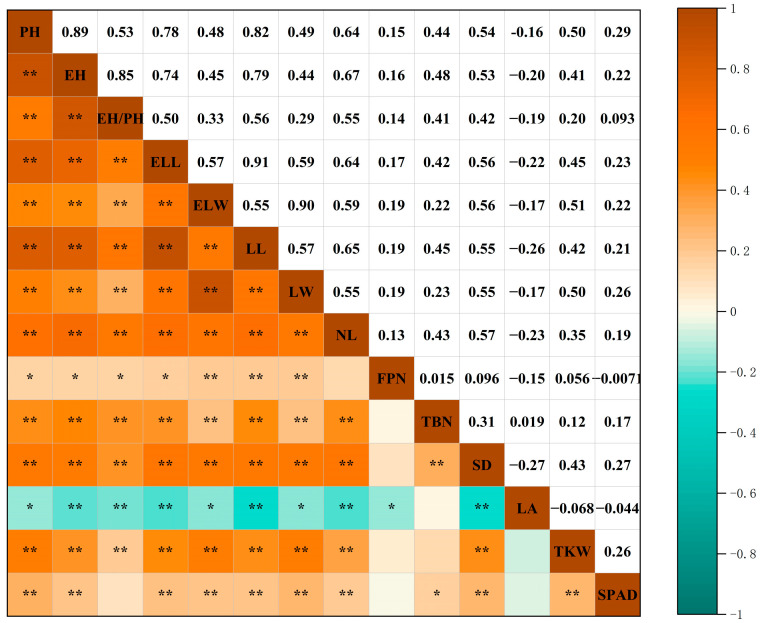
Correlation analysis of 192 maize accessions for phenotypic traits. Note: PH, EH, EH/PH, ELL, ELW, LL, LW, NL, FPN, TBN, SD, LA, TKW, and SPAD are plant height, ear height, ear height to plant height ratio, spike leaf length, spike leaf width, leaf length of upper ear, leaf width of upper ear, leaf number, effective spike, tassel branch number, stalk diameter, stem–leaf angle, thousand kernel weight, and chlorophyll content. * indicates significant correlation at 0.05 level; ** indicates significant correlation at 0.01 level.

**Figure 3 plants-13-01397-f003:**
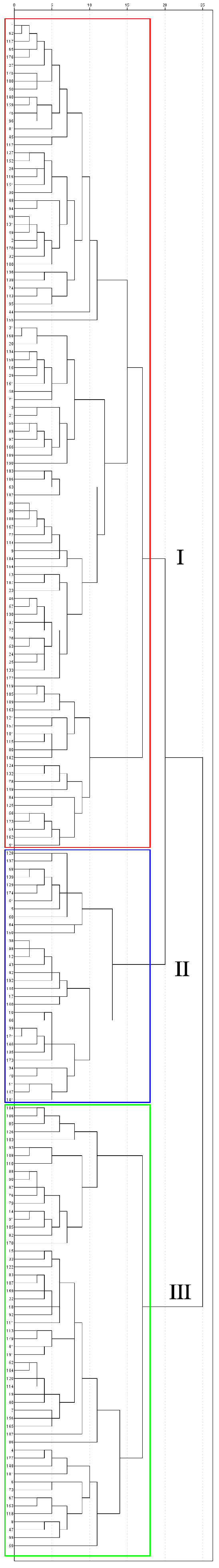
Systematic clustering of 192 accessions. Note: The red areas in the figure are for group I, the blue areas are for group II, and the green areas are for group III.

**Figure 4 plants-13-01397-f004:**
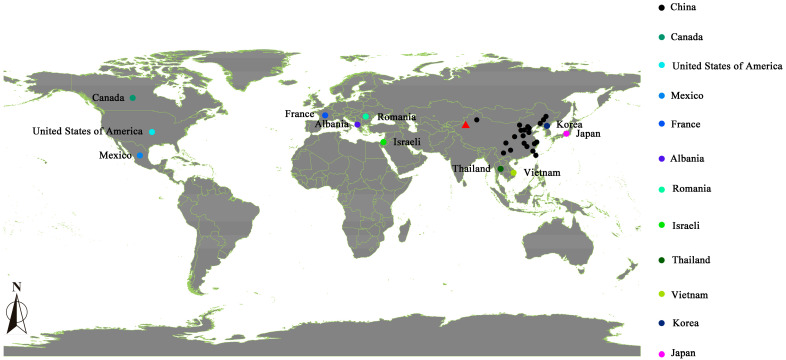
Sources of various maize accessions. Note: The red triangles represent the locations of this test in the figure; black circles represent the province of China; two accessions from this trial were from the USSR, which is not labeled on this map.

**Figure 5 plants-13-01397-f005:**
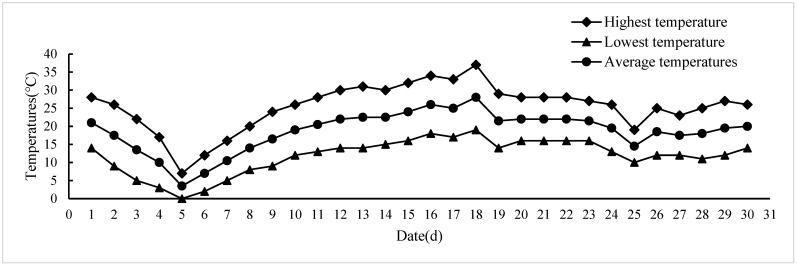
Temperature change in May 2023.

**Table 1 plants-13-01397-t001:** Diversity analysis of 192 maize accessions for 14 phenotypic traits in two years.

	2022						2023					
Traits	Min	Max	Mean	SD	CV (%)	H’	Min	Max	Mean	SD	CV (%)	H’
Plant height/cm	117.07	360.27	213.82	49.81	23.30	2.036	97.33	288.60	192.28	39.07	20.32	2.079
Ear height/cm	29.10	213.57	98.28	37.54	38.20	1.994	29.33	179.23	99.57	32.13	32.27	2.063
Ear height to plant height ratio/%	21.90	74.48	44.91	9.34	20.78	2.079	18.57	83.29	51.19	10.36	20.24	2.053
Spike leaf length/cm	46.33	105.20	77.28	12.91	16.71	2.092	33.60	168.27	73.03	13.72	18.78	1.959
Spike leaf width/cm	5.67	11.33	8.37	1.32	15.73	2.078	5.03	12.23	8.58	1.33	15.55	2.055
Leaf length of upper ear/cm	42.60	109.70	76.18	13.72	18.02	2.069	30.70	104.00	70.77	11.76	16.62	2.069
Leaf width of upper ear/cm	5.17	11.47	8.31	1.20	14.48	2.093	4.70	11.60	8.51	1.30	15.30	2.066
Leaf number/piece	7.67	22.67	13.94	2.51	17.99	2.032	7.33	18.67	13.15	2.25	17.14	2.080
Effective spikes	0.77	1.08	0.97	0.05	5.60	2.023	0.88	1.07	0.98	0.04	3.78	2.020
Tassel branch number	5.00	34.67	17.76	5.70	32.11	2.069	5.33	31.00	16.66	5.06	30.37	2.086
Stem diameter/mm	13.22	35.76	24.17	3.88	16.06	2.058	15.71	38.02	26.22	4.53	17.28	2.081
Stem–leaf angle	21.67	73.67	41.08	8.53	20.76	1.993	18.67	57.33	35.42	6.97	19.68	2.049
Thousand kernel weight/g	59.24	433.10	220.50	63.92	28.99	2.058	71.57	502.88	218.55	63.30	28.96	2.005
Chlorophyll content	22.00	59.43	41.06	7.00	17.04	2.043	21.83	65.97	43.97	8.72	19.84	2.063

Note: H’ represents the Shannon–Weaver diversity index.

**Table 2 plants-13-01397-t002:** Analysis of variance for 14 phenotypic traits in 2 years.

Source of Variation	Plant Height	Ear Height	Ear Height to Plant Height Ratio	Spike Leaf Length	Spike Leaf Width	Leaf Length of Upper Ear	Leaf width of Upper Ear	Leaf Number	Effective Spikes	Tassel Branch Number	Stem Diameter	Stem–Leaf Angle	Thousand Kernel Weight	Chlorophyll Content
Year	900.67 **	6.13	564.22 **	125.02 **	24.79 **	229.27 **	17.82 **	183.71 **	144.11 **	94.19 **	152.34 **	445.87 **	4.33	208.33 **
Germplasm	57.06 **	72.86 **	22.13 **	19.23 **	14.85 **	21.27 **	9.90 **	26.64 **	2.54 **	32.24 **	8.50 **	9.20 **	64.37 **	16.95 **
Year * Germplasm	23.99 **	20.68 **	6.87 **	6.35 **	5.32 **	5.49 **	4.26 **	8.55 **	2.00 **	14.39 **	4.92 **	8.40 **	32.06 **	14.81 **

Note: * indicates significant at the 0.05 level; ** indicates significant at the 0.01 level.

**Table 3 plants-13-01397-t003:** Eigenvalues and eigenvector descriptions of the first three principal components of the 14 phenotypic traits.

Traits	PC1	PC2	PC3
Plant height/cm	0.78	0.43	0.04
Ear height/cm	0.91	0.26	0.13
Ear height to plant height ratio/%	0.82	0.03	0.19
Spike leaf length/cm	0.70	0.50	0.16
Spike leaf width/cm	0.23	0.83	0.24
Leaf length of upper ear/cm	0.76	0.44	0.20
Leaf width of upper ear/cm	0.22	0.85	0.21
Leaf number/piece	0.65	0.44	0.18
Effective spikes	0.06	0.07	0.68
Tassel branch number	0.69	0.06	−0.24
Stalk diameter/mm	0.44	0.58	0.16
Stem–leaf angle	−0.13	−0.09	−0.67
Thousand kernel weight/g	0.18	0.72	−0.06
chlorophyll content	0.12	0.49	−0.34
Eigenvalue	4.41	3.40	1.37
Contribution rate/%	31.53	24.29	9.75
Cumulative contribution/%	31.53	55.82	65.57

Note: PC1, PC2, and PC3 are the first principal components, second principal components, and third principal components.

**Table 4 plants-13-01397-t004:** Quantitative characteristics of three groups of maize accessions.

Traits	I		II		III	
	Mean	CV (%)	Mean	CV (%)	Mean	CV (%)
Plant height/cm	198.50	13.11	153.83	7.99	239.61	11.32
Ear height/cm	96.82	22.90	60.05	20.16	125.03	21.42
Ear height to plant height ratio/%	48.63	16.11	39.01	17.35	52.14	13.79
Spike leaf length/cm	75.12	11.26	60.53	11.89	83.58	12.15
Spike leaf width/cm	8.38	11.31	7.37	15.00	9.29	9.16
Leaf length of upper ear/cm	73.61	11.97	58.42	12.06	81.83	10.57
Leaf width of upper ear/cm	8.35	10.47	7.32	13.20	9.14	8.64
Leaf number/piece	13.42	12.84	11.42	16.61	15.01	10.61
Effective spike	0.98	3.62	0.95	4.19	0.98	4.17
Tassel branch number	17.35	24.38	14.22	36.61	18.66	19.79
Stalk diameter/mm	25.14	10.85	21.69	15.26	27.29	9.97
Stem–leaf angle/θ	38.10	15.14	39.44	14.58	37.85	14.01
Thousand kernel weight/g	206.26	16.35	160.93	20.14	277.63	10.74
Chlorophyll content	42.03	14.01	39.33	11.28	45.23	11.30

**Table 5 plants-13-01397-t005:** Codes, source and type of accessions for each group.

Group	Germplasm Code	Source	Types of Germplasm
I (104 copies)	1, 2, 3, 9, 13, 16, 20, 21, 23, 24, 25, 26, 27, 28, 29, 30, 31, 32, 35, 36, 37, 44, 45, 46, 48, 49, 50, 51, 53, 54, 55, 56, 57, 58, 62, 63, 65, 69, 71, 72, 74, 75, 77, 78, 80, 81, 84, 89, 94, 95, 96, 97, 100, 101, 109, 112, 115, 116, 117, 119, 121, 123, 124, 125, 127, 130, 131, 132, 133, 134, 136, 138, 140, 142, 143, 144, 149, 151, 152, 154, 155, 157, 158, 159, 161, 162, 163, 166, 167, 168, 170, 172, 175, 176, 180, 182, 183, 184, 185, 186, 187, 188, 189, 190	Beijing province, Tianjin province, Hebei province, Liaoning province, Inner Mongolia Autonomous Region, Jilin province, Heilongjiang province, Zhejiang province, Fujian province, Jiangxi province, Shandong province, Henan province, Hubei province, Sichuan province, Guizhou province, Yunnan province, Shaanxi province, Xinjiang Uighur Autonomous Region, Foreign countries	Domestic local varieties (79) Foreign varieties (11) Domestic Autogamous Varieties (3) Foreign Self-inherited Lines (11)
II (32 copies)	5, 10, 11, 12, 17, 34, 38, 39, 42, 43, 60, 61, 64, 66, 68, 70, 98, 128, 129, 135, 137, 139, 141, 145, 146, 147, 150, 160, 171, 173, 174, 192	Hebei province, Inner Mongolia Autonomous Region, Heilongjiang province, Jilin province, Shandong province, Henan province, Sichuan province, Xinjiang Uighur Autonomous Region, Beijing province, Fujian province, foreign countries, Shanghai province	Domestic local varieties (27) Foreign varieties (1) Domestic autogamous lines (3) Domestic population (1)
III (56 copies)	4, 6, 7, 8, 14, 15, 18, 19, 22, 33, 40, 41, 47, 52, 59, 67, 73, 76, 79, 82, 83, 85, 86, 87, 88, 90, 91, 92, 93, 99, 102, 103, 104, 105, 106, 107, 108, 110, 111, 113, 114, 118, 120, 122, 126, 148, 153, 156, 164, 165, 169, 177, 178, 179, 181, 191.	Hebei province, Shanxi province, Liaoning province, Jilin province, Zhejiang province, Jiangxi province, Shandong province, Henan province, Hubei province, Guizhou province, Yunnan province, Shaanxi province, Xinjiang Uighur Autonomous Region, Foreign countries	Domestic local varieties (29) Foreign varieties (2) Foreign inbred lines (3)

**Table 6 plants-13-01397-t006:** Comparison of temperature and precipitation for April–October 2022–2023.

Vintages	Months	April	May	June	July	August	September	October
2022	Monthly minimum temperature/°C	9.0	16.0	17.0	19.0	15.0	13.0	3.0
	Monthly maximum temperature/°C	26.0	31.0	33.0	34.0	29.0	31.0	20.0
	Average temperature/°C	17.5	23.5	25.0	26.5	22.0	22.0	11.5
	Precipitation/mm	0.0	0.0	0.0	0.5	31.7	0.0	0.0
2023	Monthly minimum temperature/°C	7.0	12.0	18.0	20.0	18.0	13.0	8.0
	Monthly maximum temperature/°C	22.0	26.0	33.0	35.0	34.0	29.0	26.0
	Average temperature/°C	14.5	19.0	25.5	27.5	26.0	21.0	17.0
	Precipitation/mm	3.7	2.2	0.7	6.1	9.2	2.1	0.0

**Table 7 plants-13-01397-t007:** Groupification of trait observations.

Hierarchy	Trait Observations	Hierarchy	Trait Observations
1	$X_{1}$ < $\;\bar{X}$ − 2σ	6	$\bar{X}\;$ ≤ $\;X_{6}$ < $\;\bar{X}$ + 0.5σ
2	$\bar{X}$ − 2σ ≤${\;X}_{2}$ < $\;\bar{X}$ − 1.5σ	7	$\bar{X}\;$ + 0.5σ ≤$\;X_{7}$ < $\;\bar{X}$ + σ
3	$\bar{X}$ − 1.5σ ≤$\;X_{3}$ < $\;\bar{X}$ − 1σ	8	$\bar{X}$ + 1σ ≤$\;X_{8}$ < $\;\bar{X}$ + 1.5σ
4	$\bar{X}$ − σ ≤$\;X_{4}$ < $\bar{\;X}$ − 0.5σ	9	$\bar{X}$ + 1.5σ ≤${\;X}_{9}$ < $\;\bar{X}$ + 2σ
5	$\bar{X}$ − 0.5σ ≤$\;X_{5}$ < $\;\bar{X}$	10	$X_{10}$ ≥ $\bar{\;X}$ + 2σ

Note: ‘*X*’ denotes the trait value, ‘$\bar{X}$’ denotes the overall trait mean, and ‘σ’ denotes the trait standard deviation.

## Data Availability

Data is contained within the article.

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
