# Peer review of "Comprehensive Evaluation and Selection of 192 Maize Accessions from Different Sources"

_plants, 2024, doi:10.3390/plants13101397_

Round 1

Reviewer 1 Report

Comments and Suggestions for Authors

Dear Authors,

I had the opportunity to read and review the manuscript entitled „Comprehensive evaluation and selection of 192 maize germplasm resources from different sources”.

The study aimed to analyze the diversity and genetic changes of agronomic traits in maize germplasm resources, particularly focusing on those suitable for cultivation in the Aral region in Xinjiang and similar climatic areas. To achieve this aim, the researchers conducted a comprehensive analysis using various methods such as correlation analysis, principal component analysis, affiliation function method, stepwise regression, cluster analysis, and GGE biplots.

In my opinion, the manuscript is interesting and in line with the journal's scope: taking into account site-specific conditions, this study significantly contributes to the efficient use of existing germplasm resources, aid in the selection and breeding of new maize varieties, and provide valuable insights for maize cultivation in specific regions like the Aral region in Xinjiang. My review below suggests some improvements.

Title: Informative and consistent with the study's content.

Abstract: The abstract is clear and reasonable.

1. Introduction: The Introduction provides a clear purpose and rationale for the study. The primary objective is to systematically evaluate the diversity of maize germplasm resources and their growth characteristics in the Aral region of Xinjiang. The authors aim to achieve this by studying 14 agronomic traits of 192 maize germplasm resources over two years, from 2022 to 2023. They intend to construct comprehensive evaluation indices using multivariate statistical methods and analyze the stability of these indices using a GGE biplot.

The introduction effectively contextualizes the significance of maize cultivation in China, particularly in Xinjiang, and the importance of understanding the adaptability and stability of maize germplasm resources in diverse ecological zones. It highlights the need for comprehensive analyses of agronomic traits to support germplasm screening, identification of superior genes, and breeding efforts.

Furthermore, the introduction references previous studies to demonstrate the existing knowledge gap in the specific context of maize germplasm resources in Xinjiang. It acknowledges the limited research on screening maize germplasm resources in Xinjiang and emphasizes the importance of multi-year assessments for reliable evaluations.

Overall, the purpose and rationale for the article are clearly stated in the introduction, and the objectives outlined align well with the context and research gap identified.

2. Results:

The article is completely illogical in its structure. The ’Results’ section contains the analysis of the distribution pattern of 14 agronomic traits between the two years and the diversity analysis of agronomic traits.

Different varieties showed significant differences in genetic traits based on the results of an ANOVA. To detect varieties with significant differences, I miss performing post hoc tests.

Instead of 'Correlation analysis of maize germplasm resources for various agronomic traits' the correct one is: 'Correlation analysis of various agronomic traits for 192 maize germplasm resources'.

The principal component structure is not clear enough, there are many cross-loadings which means that an observed variable (trait) has a high loading (correlation) on more than one principal component (e.g. Spike leaf width, Leaf width of upper ear, Thousand kernel weight, Chlorophyll content). It would have been appropriate to use rotation (e.g. Varimax).

The quality of GGE biplots based on principal component analysis and cluster analysis should be improved.

3. Discussion

The article's findings are compared to those of other authors, demonstrating a comparative context in which the research results. The article discusses similarities and differences between the current study's findings and those of previous research, providing a broader perspective on the significance of the research results.

For example, in the section discussing the diversity of agronomic traits in maize germplasm resources, the study's findings are compared to those of other researchers. The authors note similarities in the findings regarding the rich genetic diversity of maize germplasm resources but also acknowledge differences that may be attributed to variations in environmental conditions, climate, cultivation methods, and germplasm resource origins.

Overall, the article effectively uses comparisons with other authors' findings to demonstrate the importance of the research results.

4. Material and Methods

Data collection methods and data analysis are clearly explained.

5. Conclusions

This section briefly summarizes the research findings.

Author Response

Thank you for taking time out of your busy schedule to revise my paper

Point 1

Different varieties showed significant differences in genetic traits based on the results of an ANOVA. To detect varieties with significant differences, I miss performing post hoc tests.

Response 1

Thank you for your advice on my paper. I conducted ANOVA and post hoc test analysis, and the results of my post hoc test are in the attachment. Due to the large amount of materials required in the paper, the analysis results cannot be well presented in the form of tables in the paper or organized into attachments. I can only provide the results after its analysis is completed. I am sorry for this, and I am sorry that my paper cannot present this part of the content.

Point 2

Instead of 'Correlation analysis of maize germplasm resources for various agronomic traits' the correct one is: 'Correlation analysis of various agronomic traits for 192 maize germplasm resources'.

Reponse 2

My correlation analysis was performed on the basis of 192 maize accessions.

Point 3

The principal component structure is not clear enough, there are many cross-loadings which means that an observed variable (trait) has a high loading (correlation) on more than one principal component (e.g. Spike leaf width, Leaf width of upper ear, Thousand kernel weight, Chlorophyll content). It would have been appropriate to use rotation (e.g. Varimax).

Response 3

Thank you for your suggestion, and your opinion makes my analysis clearer. For principal component analysis, I used the rotation method you recommended to reanalyze and express the results after rotation, and the changes have been presented in the article (Line 147-160)

Point 4

The quality of GGE biplots based on principal component analysis and cluster analysis should be improved.

Response 4

Thank you for your suggestions. I corrected the GGE double map quality again to improve the picture quality (Fingure 4 and 5).

Your affirmation of my paper has greatly encouraged me. Thank you very much for your suggestions taken out of your busy schedule. Your suggestions will make my paper more perfect. Thank you very much for your suggestions。

Reviewer 2 Report

Comments and Suggestions for Authors

The manuscript “Comprehensive evaluation and selection of 192 maize germplasm resources from different sources” was professionally conducted. It merits publication in the Plants journal following revisions. Suggestions for the authors to consider are outlined below.

Line 50: … diffrent …

Suggestion: … different …

Lines 58-63: Among the various statistical methods, principal component analysis, cluster analysis, and GGE biplot are important research methods. In crop breeding, principal component analysis has been widely used for the comprehensive analysis of germplasm resources, trait relationships, and other related studies[11,12], cluster analysis is often used in species grouping, crop planting zoning, and other research[13,14], The GGE biplot is not limited to analyzing the results of regional tests.

Suggestion: Among the various statistical methods, principal component analysis (PCA), cluster analysis, and GGE biplot stand out as crucial research tools. In crop breeding, PCA finds extensive application for the comprehensive analysis of germplasm resources and trait relationships [11,12], while cluster analysis is frequently employed in species grouping and crop planting zoning [13,14]. The utility of GGE biplot extends beyond the analysis of regional test results.

Lines 66-68: Multivariate statistical analyses based on multivariate analysis have been widely used in the study of genetic patterns, relationships, and interrelation ships among traits [16,17,18].

Suggestion: Multivariate statistical analyses have been extensively employed in the study of genetic patterns, relationships, and interrelationships among traits [16,17,18].

Lines 209-211: The KMO and Bartlett tests (KMO=0.805>0.65, P<0.01) indicated a high degree of correlation among each factor, while the results of principal component analysis yielded favorable outcomes[37].

Suggestion: What do you mean by “yielded favorable outcomes”?

Lines 613-614: The field was managed in the same way as the field.

Suggestion: The word “field” is repeated.

Line 624: … continental arid desert climate63, …

Suggestion: Remove the “63” number.

Line 625: … ≥10°C is 4541.4°C, …

Suggestion: Question: 4541.4 days instead of 4541.4oC?

Lines 625-626: The city benefits from an abundance of light and heat resources, but occasional dust storms can occur.

Suggestion: The city benefits from an abundance of light and heat, but occasional dust storms can occur.

Lines 630-631: irrigation64.

Suggestion: Remove the “64” number.

Line 655: … spikes arerudimentary, …

Suggestion: … spikes are rudimentary, …

Line 655: … growth process did not stopped …

Suggestion: … growth process did not stop …

Line 657: C: The female spikelets not develop, …

Suggestion: C: The female spikelets have not developed, …

Line 658-659: The development process stopped after the seedling stage, and the plant no longer grows and develops.

Suggestion: The development process stopped after the seedling stage, and the plant no longer exhibits growth or further development.

Comment about references: Excessive number of references.

Comments on the Quality of English Language

The manuscript needs minor English language revision.

Author Response

I have revised the words and sentences you have given, and retained the traces of modification. I feel very sorry for your trouble to lack my English ability. Your advice makes my paper avoid a lot of unclear expression and improper wording or spelling errors, make my paper sentences more correct, avoid many mistakes, and also make my content more perfect. Thank you very much for your advice

Point 1

Line 50: … diffrent …

Suggestion: … different …

Response 1

agree. I have revised it (Line 45).

Point 2

Lines 58-63: Among the various statistical methods, principal component analysis, cluster analysis, and GGE biplot are important research methods. In crop breeding, principal component analysis has been widely used for the comprehensive analysis of germplasm resources, trait relationships, and other related studies[11,12], cluster analysis is often used in species grouping, crop planting zoning, and other research[13,14], The GGE biplot is not limited to analyzing the results of regional tests.

Suggestion: Among the various statistical methods, principal component analysis (PCA), cluster analysis, and GGE biplot stand out as crucial research tools. In crop breeding, PCA finds extensive application for the comprehensive analysis of germplasm resources and trait relationships [11,12], while cluster analysis is frequently employed in species grouping and crop planting zoning [13,14]. The utility of GGE biplot extends beyond the analysis of regional test results.

Response 2

agree. I have revised it according to your recommendations (Line 49-52).

Point 3

Lines 66-68: Multivariate statistical analyses based on multivariate analysis have been widely used in the study of genetic patterns, relationships, and interrelation ships among traits [16,17,18].

Suggestion: Multivariate statistical analyses have been extensively employed in the study of genetic patterns, relationships, and interrelationships among traits [16,17,18].

Response 3

agree. Revised as per your recommendation (Line 54-55)

Point 4

Lines 209-211: The KMO and Bartlett tests (KMO=0.805>0.65, P<0.01) indicated a high degree of correlation among each factor, while the results of principal component analysis yielded favorable outcomes[37].

Suggestion: What do you mean by “yielded favorable outcomes”?

Response 4

Thank you for your advice. My explanation is as follows: KMO and Bartlett tests are required before PCA, and the data is meaningful only when KMO is> 0.65 and is significant. I changed 'while the results of principal component analysis yielded favorable outcomes' to 'principal component analysis can be performed on the data' (Line 149-151). Maybe my grammar problems and inaccurate word use have caused the sentence ambiguity, for which I am sorry.

Point 5

Lines 613-614: The field was managed in the same way as the field.

Suggestion: The word “field” is repeated.

Response 5

Sorry, because ambiguity ambiguity ambiguity my ambiguity. I changed “The field was managed in the same way as the field.” to “Corn growth at the trial site is managed in the same way as in the field.”(Line 434). In order to better express what I mean.

Point 6

Line 624: … continental arid desert climate63, …

Suggestion: Remove the “63” number.

Reponse 6

I changed ‘63’to‘[63]’,Because here I cited the literature(Lin 442). I am very sorry, due to my negligence, I did not add the symbol, creating an ambiguity.

Point 7

Line 625: … ≥10°C is 4541.4°C, …

Suggestion: Question: 4541.4 days instead of 4541.4℃?

Response 7

It selected 4541.4℃。In addition, I changed”effective cumulative temperature” to “effective accumulated temperature”(Line 442).

Point 8

Lines 625-626: The city benefits from an abundance of light and heat resources, but occasional dust storms can occur.

Suggestion: The city benefits from an abundance of light and heat, but occasional dust storms can occur.

Response 8

Agree. I have revised it according to your opinion (Line 443-444).

Point 9

Lines 630-631: irrigation64.

Suggestion: Remove the “64” number.

Response 9

I changed “64” to ‘[64]’,Because here I cited the literature(Lin 447). I am very sorry, due to my negligence, I did not add the symbol, creating an ambiguity.

Point 10

Line 655: … spikes arerudimentary, …

Suggestion: … spikes are rudimentary, …

Response 10

Agree. I revised, following your recommendations(Line 463)

Point 11

Line 655: … growth process did not stopped …

Suggestion: … growth process did not stop …

Response 11

Agree. I revised, following your recommendations(Line 465-466)

Point 12

Line 657: C: The female spikelets not develop, …

Suggestion: C: The female spikelets have not developed, …

Response 12

Agree. I revised, following your recommendations(Line 465-466)。

Point 13

Line 658-659: The development process stopped after the seedling stage, and the plant no longer grows and develops.

Suggestion: The development process stopped after the seedling stage, and the plant no longer exhibits growth or further development.

Response 13

Agree. I revised, following your recommendations(Line 467)

Point 14

Comment about references: Excessive number of references.

Response 14

Agree. I have deleted some parts of the literature.

Reviewer 3 Report

Comments and Suggestions for Authors

Thank you for this interesting study. I have some comments:

Could you please review in the phenotypic composite value the following comments?

"Composite indicator weights: Wi =P /ii ,i=1,2,3,... ,n                   (2)

Where Wi is the weight of the ith composite indicator among all composite indicators, and Pi is the contribution rate of the ith composite indicator of each variety." Please, define P.

In 6íóPrincipal component análisis.xlsx is not defined how you calculated the weighting factor (Sheet aggregate score). I understood that is derived from the percentage of variance explained by the three first PCs, is that correct? Could you please include it in the Appendix Table?

Percentage
of variance

cumulative %

47.108

47.108

10.117

57.224

8.348

65.573

Could you discuss better the “ten indices could serve as reliable indicators for evaluating maize germplasm comprehensively” when with 3 you are over the 90% of the total explained variance? if I understood the stepwise regression analysis correctly.

Please, complete the titles of the Tables and Figures with a definition of the content.

Please, review the format of the manuscript, it presents defects in the pdf.

Please, improve quality of Figure 3, 4 (pdf version).

Line 614: “The field was managed in the same way as the field.” ???????

How many repetitions per trait did you collect by year and accession? It seems 3 in the Suppl. Table. I did not find it in Methods.

Could you include a map with the origin of the accessions evaluated and the place of the trials?

Author Response

Thank you very much for taking time out of your busy schedule for your valuable advice. I am very grateful.

Point 1

"Composite indicator weights: Wi =P /ii ,i=1,2,3,... ,n                   (2)”

Where Wi is the weight of the ith composite indicator among all composite indicators, and Pi is the contribution rate of the ith composite indicator of each variety." Please, define P.

Response 1

I'm very sorry, this is a personal carelessness.As for the 'formula 2' in the first draft, the formula input is incorrectly. There is no 'P' indicator.The correct formula shall be ” Wi=Pi/i”.(Line 518).I am sorry for the error in my negligence formula.

Point 2

In “6.Principal component análisis.xlsx ”is not defined how you calculated the weighting factor (Sheet aggregate score). I understood that is derived from the percentage of variance explained by the three first PCs, is that correct? Could you please include it in the Appendix Table?

Response 2

As for the weight calculation of the "formula", your understanding is correct. The weight calculation is calculated based on the contribution rate of the three principal components, and I have presented it as an attached table(Line 520).

Point 3

Could you discuss better the “ten indices could serve as reliable indicators for evaluating maize germplasm comprehensively” when with 3 you are over the 90% of the total explained variance? if I understood the stepwise regression analysis correctly.

Response 3

I chose model 10 because the correlation coefficient and R2 of the equation are close to 1, while the correlation coefficient and R2 of model 11,12,13,14 are too close to 1 to have no meaning of choice. Model selection is subjective, so it unavoidable.

Point 4

Please, complete the titles of the Tables and Figures with a definition of the content.

Response 4

Agree. I have defined the content both in the tables and in the pictures(figure 1(Line101-102),table 4,table 9)

In Figure 1, I illustrate the coordinates, it is“The trait value is represented on the horizontal axis, and the amount of germplasm in the interval for the trait is represented on the vertical axis.”;

I have annotated it in Table 4, it is “Note: Ⅰ, Ⅱ, Ⅲ represent group Ⅰ, group Ⅱ, and group Ⅲ. The following table is the same.”;

In Table 9 I illustrate some of the symbols in the table, it is“Note: 'X' denotes the trait value, '' denotes the overall trait mean, and 'σ' denotes the trait standard deviation.”。

Point 5

Please, review the format of the manuscript, it presents defects in the pdf.

Response 5

The manuscript format was corrected according to the journal requirements and in combination with the published manuscripts of the journal。

Point 6

Please, improve quality of Figure 3, 4 (pdf version).

Response 6

Agree. I have strengthened the quality of the picture.

Point 7

Line 614: “The field was managed in the same way as the field.” ???????

Response 7

I changed “The field was managed in the same way as the field.” to “Corn growth at the trial site is managed in the same way as in the field.”(Line 434)

Point 8

How many repetitions per trait did you collect by year and accession? It seems 3 in the Suppl. Table. I did not find it in Methods.

Response 9

Agree. I will explain it clearly in the first sentence of "4.5. Data analysis".You are right, I really didn't write clearly that the data is several repetitions, and my data is indeed three repetitions(Line 502)。

Point 9

Could you include a map with the origin of the accessions evaluated and the place of the trials?

Response9

Agree. I have added it in the text by following your suggestion.Figure of the position in the text "Line 424". The way of making the map and the description is in "Line 564-567"

Reviewer 4 Report

Comments and Suggestions for Authors

REVIEW
Comprehensive evaluation and selection of 192 maize germplasm resources from different sources

Mengting Hu, Huijuan Tian, Kaizhi Yang, Shuqi Ding, Ying Hao, Ruohang Xu, Fulai Zhang, Hong Liu, Dan Zhang

1)_related sectors[1]. With the continuous promotion of maize cultivation in China, a series 40 of unique germplasm resources have been gradually accumulated. These resources con- 41

REV A series is singular therefore …resources has been…..

2) area for maize seed production has reached approximately 1.4 million mu, and the maize 44

Do not use mu use hecatres

3) yield is much higher than the national average[4,5]. However, the diverse climatic envi- 45

You are surely mixing up yield of seed corn (F1 hybrid seed for planting) with grain (F2 seed harvested from F1 hybrids)—this comparison is not valid—compare apples with apples not apples with oranges.

4) The performance of agronomic traits varies due to differences in climatic 52

conditions, soil type, fertility, and cultivation practices between years. The stability of 53

You need to repeat in this sentence the genetic effects or rewrite the first two sentences of this para eg

The diversity of phenotypic traits, including agronomic traits, is a result of both genetic and environmental diversity, including climatic 52

conditions, soil type, fertility, and cultivation practices between years and locations.

5) A comprehensive analysis of agronomic traits in maize 54

germplasm resources is essential for understanding their genetic background and comprehensive performance, and provide theoretical support for germplasm screening and the identification of superior genes, thereby offering a theoretical basis for germplasm innovation and the breeding of superior varieties

YES BUT  Examine very carefully your methods—to what extent do the methods you have employed actually provide information that reveals genetic information?  As written you overstate what this paper provides:

Rewrite as follows:

germplasm resources is an important prerequisite for subsequent studies directed toward developing an understanding their genetic background

6) There is a serious problem calling the 14 traits you recorded as agronomic traits. Only one trait measured (thousand kernel weight) has any real relevance to an agronomic  trait. Agronomic traits include yield, maturity, date of pollen shed, date of silking, standability, insect resistances, disease resistances. You recorded none of these. The study is neither comprehensive nor is it of agronomic traits-phenotypic traits yes, but certainly not agronomic,

7) Currently, there have been more studies on the morphological identification and as- 69

Rewrite:

Several studies have been reported on the morphological identification and as

8) Related to comment 5)

; Bedoya et al. conducted a study on the genetic diversity and population structure of 194 local maize populations from countries in the United States and the Caribbean region. The study revealed that the maize populations in this region display a rich genetic diversity. Additionally, the study identified three major groups of maize germplasm[23];

Yes BUT Bedoya et al al used genetic data—ie Simple Sequence repeats so they can legitimately speak to  genetic diversity.

9) Tanavar M et al. evaluated 13 maize inbred lines based on morphological traits and found that the outcomes of the principal component analysis and the cluster analysis were nearly identical[26];

This result is expected—how does it relate to your study?

10) Meng Zuqing et al. conducted multi-year replicated trials to assess the phenotypic diversity of 179 local maize varieties in Tibet. . This study will serve as a foundation for the protection and development of local maize varieties in Tibet[28];

RECHECK this paper--

The closest publication I could find by these authors was

 Meng, Z., F. Song and T. Liu, 2018. Genetic diversity and genetic structure analysis of maize (Zea mays) landraces in Tibet. Int. J. Agric.

Biol., 20: 791−798

Genetic Diversity and Genetic Structure Analysis of Maize (Zea mays)

Landraces in Tibet

They made no such statement as “This study will serve as a foundation for the protection and development of local maize varieties in Tibet[28];” Are you quoting their paper if so OK OR are you making this assertion yourselves—NOT OK.

11) This study aimed to systematically evaluate the diversity of maize germplasm resources collected and their growth habits and characteristics in the Arar region of Xinjiang. Fourteen traits of 192 maize germplasm resources introduced over a two-year period were studied from 2022 to 2023. Comprehensive evaluation indices for maize germplasm resources were constructed using multivariate statistical methods. The stability of these indexes was analyzed using a GGE biplot to identify maize germplasm with excellent performance. This provides a theoretical basis for innovation in maize germplasm, the selection of new varieties, and breeding.

IN your study: Two years is insufficient to properly study phenotypic diversity and stability

As mentioned under 6) only one trait- thousand grain weight can be considered as agronomic, None of the other characteristics reported  are agronomically important. The results of this study provide no  basis of support for selection and further breeding.

Using a multitude of analytical approaches does NOT make up for a lack of diversity in years and locations employed.

Using one location is insufficient to determine phenotypic diversity and plasticity

12) Materials and Methods section should come before results!

13) Do NOT lump all 192 accessions as germplasm resources. Yes the local varieties can be considered as “germplasm resources” BUT all the accessions that are either inbred lines or hybrids should be classified accordingly. They  are essentially check accessions that have been or can be reported upon widely. You need to reduce the number of germplasm resources accordingly. Cite where the accessions are listed in the text.

Author Response

Thank you very much for taking time out of your busy schedule for your valuable advice. I am very grateful+

Point 1

related sectors[1]. With the continuous promotion of maize cultivation in China, a series 40 of unique germplasm resources have been gradually accumulated. These resources con- 41

REV A series is singular therefore …resources has been…..

Response 1

Agree. Following your suggestion, I have revised the statement(Line 39-40)。

Point 2

area for maize seed production has reached approximately 1.4 million mu, and the maize 44

Do not use mu use hecatres

Response 2

Agree. You should indeed use "hecatres"

Point 3

yield is much higher than the national average[4,5]. However, the diverse climatic envi- 45

You are surely mixing up yield of seed corn (F1 hybrid seed for planting) with grain (F2 seed harvested from F1 hybrids)—this comparison is not valid—compare apples with apples not apples with oranges.

Response 3

Agree. There is indeed an ambiguity in this sentence. After careful thinking, I think the relevant statement you proposed can be deleted, but the sentence has little effect in the text.

Point 4

The performance of agronomic traits varies due to differences in climatic

conditions, soil type, fertility, and cultivation practices between years. The stability of

You need to repeat in this sentence the genetic effects or rewrite the first two sentences of this para eg

The diversity of phenotypic traits, including agronomic traits, is a result of both genetic and environmental diversity, including climatic ,conditions, soil type, fertility, and cultivation practices between years and locations.

Response 4

Agree. Your question good the point. I have revised this content by following your suggestions(Line 46-47)

Point 5

A comprehensive analysis of agronomic traits in maize

germplasm resources is essential for understanding their genetic background and comprehensive performance, and provide theoretical support for germplasm screening and the identification of superior genes, thereby offering a theoretical basis for germplasm innovation and the breeding of superior varieties

YES BUT  Examine very carefully your methods—to what extent do the methods you have employed actually provide information that reveals genetic information?  As written you overstate what this paper provides:

Rewrite as follows:

germplasm resources is an important prerequisite for subsequent studies directed toward developing an understanding their genetic background

Response 5

Agree. Your suggestion is very pertinent, and I have revised it according to your suggestion(Line 47-49)

Point 6

There is a serious problem calling the 14 traits you recorded as agronomic traits. Only one trait measured (thousand kernel weight) has any real relevance to an agronomic  trait. Agronomic traits include yield, maturity, date of pollen shed, date of silking, standability, insect resistances, disease resistances. You recorded none of these. The study is neither comprehensive nor is it of agronomic traits-phenotypic traits yes, but certainly not agronomic,

Response 6

Your advice hit the nail on the head. Due to my ambiguity of the concept, I caused the concept error in the article, and I have changed "agronomic traits" to "phenotypic traits". Your advice gave me the answer.

Point 7

Currently, there have been more studies on the morphological identification and as-

Rewrite:

Several studies have been reported on the morphological identification and as

Response 7

Agree. I have revised it according to your recommendations(Line 56)

Point 8

Bedoya et al. conducted a study on the genetic diversity and population structure of 194 local maize populations from countries in the United States and the Caribbean region. The study revealed that the maize populations in this region display a rich genetic diversity. Additionally, the study identified three major groups of maize germplasm[23];

Yes BUT Bedoya et al al used genetic data—ie Simple Sequence repeats so they can legitimately speak to  genetic diversity.

Response 8

Agree. I am very sorry that the reference of this literature is not inappropriate in my article. I deleted the reference from the literature。

Point 9

Tanavar M et al. evaluated 13 maize inbred lines based on morphological traits and found that the outcomes of the principal component analysis and the cluster analysis were nearly identical[26];

This result is expected—how does it relate to your study?

Response 9

Agree. Your suggestion is really very pertinent. My original intention to cite the literature is that my article uses principal component and cluster analysis, and there is consistency in their results, and the purpose of quoting this article is to prove this result. After thinking over your advice, I realized that quoting this literature seems to have little use. Therefore, I removed the reference from the literature.

Point 10

local maize varieties in Tibet. . This study will serve as a foundation for the protection and development of local maize varieties in Tibet[28];

RECHECK this paper--

The closest publication I could find by these authors was“ Meng, Z., F. Song and T. Liu, 2018. Genetic diversity and genetic structure analysis of maize (Zea mays) landraces in Tibet. Int. J. Agric.Biol., 20: 791−798.”

They made no such statement as “This study will serve as a foundation for the protection and development of local maize varieties in Tibet[28];” Are you quoting their paper if so OK OR are you making this assertion yourselves—NOT OK.

Response 10

I did cite the author, the paper exists and cited correctly, This is its title and DOI.(Meng Z Q, Song F P. Phenotypic diversity and group classification of maize landraces in Tibet[J]. Journal of China Agricultural University, 2017, 22(7): 10-23.DOI:10.11841/j.issn.1007-4333.2017.07.02)。At the same time, I have downloaded this paper and will upload it in the form of an attachment.

Point11

This study aimed to systematically evaluate the diversity of maize germplasm resources collected and their growth habits and characteristics in the Arar region of Xinjiang. Fourteen traits of 192 maize germplasm resources introduced over a two-year period were studied from 2022 to 2023. Comprehensive evaluation indices for maize germplasm resources were constructed using multivariate statistical methods. The stability of these indexes was analyzed using a GGE biplot to identify maize germplasm with excellent performance. This provides a theoretical basis for innovation in maize germplasm, the selection of new varieties, and breeding.

IN your study: Two years is insufficient to properly study phenotypic diversity and stability

As mentioned under 6) only one trait- thousand grain weight can be considered as agronomic, None of the other characteristics reported  are agronomically important. The results of this study provide no  basis of support for selection and further breeding.

Using a multitude of analytical approaches does NOT make up for a lack of diversity in years and locations employed.

Using one location is insufficient to determine phenotypic diversity and plasticity

Response 11

Agree. Your words let me have a deeper understanding of the relevant knowledge. At the same time, your suggestions are also very pertinent, to make my paper more comprehensive and reasonable. This passage is indeed inappropriate in the passage. I have made the changes in the text by following your suggestion(Line 79-82).

Point 12

Materials and Methods section should come before results!

Response 12

I modified the paper format according to the requirements of the journal,I hope you excuse.

Point 13

Do NOT lump all 192 accessions as germplasm resources. Yes the local varieties can be considered as “germplasm resources” BUT all the accessions that are either inbred lines or hybrids should be classified accordingly. They  are essentially check accessions that have been or can be reported upon widely. You need to reduce the number of germplasm resources accordingly. Cite where the accessions are listed in the text.

Response 13

Thank you for your advice. The following is my description: My material type is domestic local varieties, foreign varieties, domestic inbred lines and foreign inbred lines, these variety types are established, can not be subdivided, I hope you excuse. I have marked the material name, source and type.

Round 2

Reviewer 4 Report

Comments and Suggestions for Authors

I cannot recommend publication. The study has basic fundamental problems that cannot be corrected.

You have done about every possible analysis possible on these phenotypic traits. These data CANNOT show kinship nor can they measure genetic diversity. NOR can these data provide a means to rank or identify potentially useful germplasm for further breeding. Two years is insufficient for a stability analysis. You are making WAY MORE conclusions from these analyses than is scientifically  warranted. Just show the associations DROP the evaluation sections.

1)         Abstract You still refer to 295 maize germplasm resources, change to 295 maize accessions

I would recommend you state 192 accessions and ignore the accessions with abnormal development. UNLESS there is good reason to cite those accessions with abnormal development in a table.

2)         “ A series of unique germplasm resources have gradually accumulated with the continuous promotion of maize cultivation in China.” You need to provide some indication of how this series has accumulated—by breeding, by introductions form other regions or countries—from where, mutation, selection by farmers etc etc and provide references

3)         “The aim was to understand their growth habits and characteristics and to screen out maize germplasm with better comprehensive performances. This will provide a theoretical basis for later directed breeding based on the characteristics of different maize germplasm.: THIS is a FUNDAMENTAL problem you have-POINT 6 from the previous review: There is a serious problem calling the 14 traits you recorded as agronomic traits. Only one trait measured (thousand kernel weight) has any real relevance to an agronomic trait. Agronomic traits include yield, maturity, date of pollen shed, date of silking, standability, insect resistances, disease resistances. You recorded none of these. The study is neither comprehensive nor is it of agronomic traits-phenotypic traits yes, but certainly not agronomic.

AT BEST you can state: The aim was to understand their growth habits and characteristics of several phenotypic characteristics.

4)         “Cluster analysis can be used to explore the kinship relationships between different maize germplasm resources and group them accordingly[33].: The references are not numbered so I was unable to view ref 33. HOWEVER, I disagree that cluster analysis of MORPHOLOGICAL traits can be used to investigate KINSHiP ie pedigree or genetic relationships YES cluster analysis can show associations based on the data used BUT comparisons of morphological data are notoriously unable to show kinship. 

5)         The study revealed that group II had the poorest performance based on the results of principal component and cluster analysis. I DO NOT AGREE that you can use these morphological data to rank performance as breeding materials.  You do not know the genetics of the traits.  You cannot evaluate traits in terms of useful breeding materials  based on their degree of correlation or multivariate analysis. So all the ranking of germplasm in terms of useful breeding material is not based on sound scientific principles. Sections 2.7,  2.9.1 (evaluation of utility in breeding) and 2.8 (in respect of showing kinship) are not supported by the data.

6)         Two years is insufficient for all the stability conclusions

7)         The diversity analysis results indicate that the test material exhibits rich genetic diversity, consistent with the findings of Ma Yanhua[37] and Meng Zuqing[22]. However, this differs from the results of Sirlene Viana d.F[38] and Syahruddin K[39].   NO NO NO you cannot state these data show “rch genetic diversity” First of all WHAT do you mean by “rich” ? Certainly you cannot mean valuable in terms of breeding, second morphological diversity can mean a range of genetic diversity—depending on the genetic control of traits-so you can have no idea how much genetic diversity  exists among these accessions.

8)         Effective spikes and thousand kernel weight are important factors in yield composition[45][46]. The morphology and distribution of maize leaves have a direct impact on the interception of light energy by the population canopy and the efficiency of photosynthetic utilization, which ultimately affects yield formation[47]. The study found a significant or highly significant positive correlation between leaf length and width, and chlorophyll content, effective spikes, and thousand kernel weight. Therefore, it is important to consider leaf blade size during the breeding process. The study analysed 192 maize germplasm resources, which exhibit a high level of diversity, reflecting rich genetic variation. Therefore, the traits can be targeted according to the needs of different breeding objectives.

This is all a gross over-simplification of how these traits may or may not be useful –AGAIN you cannot say “rich genetic diversity”  Traits can be targeted for selection BUT THE KEY TRAIT IS YIELD.

Comments on the Quality of English Language

I cannot recommend publication. The study has basic fundamental problems that cannot be corrected.

You have done about every possible analysis possible on these phenotypic traits. These data CANNOT show kinship nor can they measure genetic diversity. NOR can these data provide a means to rank or identify potentially useful germplasm for further breeding. Two years is insufficient for a stability analysis. You are making WAY MORE conclusions from these analyses than is scientifically  warranted. Just show the associations DROP the evaluation sections.

1)         Abstract You still refer to 295 maize germplasm resources, change to 295 maize accessions

I would recommend you state 192 accessions and ignore the accessions with abnormal development. UNLESS there is good reason to cite those accessions with abnormal development in a table.

2)         “ A series of unique germplasm resources have gradually accumulated with the continuous promotion of maize cultivation in China.” You need to provide some indication of how this series has accumulated—by breeding, by introductions form other regions or countries—from where, mutation, selection by farmers etc etc and provide references

3)         “The aim was to understand their growth habits and characteristics and to screen out maize germplasm with better comprehensive performances. This will provide a theoretical basis for later directed breeding based on the characteristics of different maize germplasm.: THIS is a FUNDAMENTAL problem you have-POINT 6 from the previous review: There is a serious problem calling the 14 traits you recorded as agronomic traits. Only one trait measured (thousand kernel weight) has any real relevance to an agronomic trait. Agronomic traits include yield, maturity, date of pollen shed, date of silking, standability, insect resistances, disease resistances. You recorded none of these. The study is neither comprehensive nor is it of agronomic traits-phenotypic traits yes, but certainly not agronomic.

AT BEST you can state: The aim was to understand their growth habits and characteristics of several phenotypic characteristics.

4)         “Cluster analysis can be used to explore the kinship relationships between different maize germplasm resources and group them accordingly[33].: The references are not numbered so I was unable to view ref 33. HOWEVER, I disagree that cluster analysis of MORPHOLOGICAL traits can be used to investigate KINSHiP ie pedigree or genetic relationships YES cluster analysis can show associations based on the data used BUT comparisons of morphological data are notoriously unable to show kinship. 

5)         The study revealed that group II had the poorest performance based on the results of principal component and cluster analysis. I DO NOT AGREE that you can use these morphological data to rank performance as breeding materials.  You do not know the genetics of the traits.  You cannot evaluate traits in terms of useful breeding materials  based on their degree of correlation or multivariate analysis. So all the ranking of germplasm in terms of useful breeding material is not based on sound scientific principles. Sections 2.7,  2.9.1 (evaluation of utility in breeding) and 2.8 (in respect of showing kinship) are not supported by the data.

6)         Two years is insufficient for all the stability conclusions

7)         The diversity analysis results indicate that the test material exhibits rich genetic diversity, consistent with the findings of Ma Yanhua[37] and Meng Zuqing[22]. However, this differs from the results of Sirlene Viana d.F[38] and Syahruddin K[39].   NO NO NO you cannot state these data show “rch genetic diversity” First of all WHAT do you mean by “rich” ? Certainly you cannot mean valuable in terms of breeding, second morphological diversity can mean a range of genetic diversity—depending on the genetic control of traits-so you can have no idea how much genetic diversity  exists among these accessions.

8)         Effective spikes and thousand kernel weight are important factors in yield composition[45][46]. The morphology and distribution of maize leaves have a direct impact on the interception of light energy by the population canopy and the efficiency of photosynthetic utilization, which ultimately affects yield formation[47]. The study found a significant or highly significant positive correlation between leaf length and width, and chlorophyll content, effective spikes, and thousand kernel weight. Therefore, it is important to consider leaf blade size during the breeding process. The study analysed 192 maize germplasm resources, which exhibit a high level of diversity, reflecting rich genetic variation. Therefore, the traits can be targeted according to the needs of different breeding objectives.

This is all a gross over-simplification of how these traits may or may not be useful –AGAIN you cannot say “rich genetic diversity”  Traits can be targeted for selection BUT THE KEY TRAIT IS YIELD.

Author Response

Thank you very much for your second suggestion, which made my paper more reasonable and reliable, and for the changes you made to my essay during your busy schedule.I have improved the quality of English in my essay. The structure of the article has been partially adjusted during the revision process.

Point 1

Abstract You still refer to 295 maize germplasm resources, change to 295 maize accessions

I would recommend you state 192 accessions and ignore the accessions with abnormal development. UNLESS there is good reason to cite those accessions with abnormal development in a table.

Response 1

Agreed. I have replaced "maize germplasm resources" with "maize accessions". I have removed the content related to the developmentally abnormal material

Point 2

A series of unique germplasm resources have gradually accumulated with the continuous promotion of maize cultivation in China.” You need to provide some indication of how this series has accumulated—by breeding, by introductions form other regions or countries—from where, mutation, selection by farmers etc etc and provide references

Response 2

Agreed. I have added 2-3 relevant literature (Lines 31-32). I will provide DOI for specific literature for your convenience.(1)DOI:10.19462/j.cnki.1671-895x.20190923.013;(2) https://doi.org/10.2135/cropsci2012.11.0645;

Point 3

“The aim was to understand their growth habits and characteristics and to screen out maize germplasm with better comprehensive performances. This will provide a theoretical basis for later directed breeding based on the characteristics of different maize germplasm.: THIS is a FUNDAMENTAL problem you have-POINT 6 from the previous review: There is a serious problem calling the 14 traits you recorded as agronomic traits. Only one trait measured (thousand kernel weight) has any real relevance to an agronomic trait. Agronomic traits include yield, maturity, date of pollen shed, date of silking, standability, insect resistances, disease resistances. You recorded none of these. The study is neither comprehensive nor is it of agronomic traits-phenotypic traits yes, but certainly not agronomic.

AT BEST you can state: The aim was to understand their growth habits and characteristics of several phenotypic characteristics.

Response 3

Agreed. Changes have been made as per your comments. Line 67-70

Point 4

“Cluster analysis can be used to explore the kinship relationships between different maize germplasm resources and group them accordingly[33].: The references are not numbered so I was unable to view ref 33. HOWEVER, I disagree that cluster analysis of MORPHOLOGICAL traits can be used to investigate KINSHiP ie pedigree or genetic relationships YES cluster analysis can show associations based on the data used BUT comparisons of morphological data are notoriously unable to show kinship.

Response 4

Agreed. I made changes to that sentence. You are indeed correct. I changed the original text to “Cluster analysis facilitates the grouping of accession metrics, enabling the classification of genetically similar accessions into homogeneous groups. This method elucidates the characteristics and relationships among taxa”.Line 159-160

Point 5

The study revealed that group II had the poorest performance based on the results of principal component and cluster analysis. I DO NOT AGREE that you can use these morphological data to rank performance as breeding materials.  You do not know the genetics of the traits.  You cannot evaluate traits in terms of useful breeding materials  based on their degree of correlation or multivariate analysis. So all the ranking of germplasm in terms of useful breeding material is not based on sound scientific principles. Sections 2.7,  2.9.1 (evaluation of utility in breeding) and 2.8 (in respect of showing kinship) are not supported by the data.

Response 5

Agreed. I have modified the content of the text. 2.7 I have deleted. The biplot was made based on cluster analysis into three categories, and only the thousand grain weight was done. I don't know if this is reasonable.

Point 6

Two years is insufficient for all the stability conclusions

Response 6

I ended up keeping the biplot plot for thousand grain weight. There is literature showing that it is possible to use biplots for stability analyses of two-year traits. (DOI:10.14083/j.issn.1001-4942.2023.02.005.  )

Point 7

The diversity analysis results indicate that the test material exhibits rich genetic diversity, consistent with the findings of Ma Yanhua[37] and Meng Zuqing[22]. However, this differs from the results of Sirlene Viana d.F[38] and Syahruddin K[39].   NO NO NO you cannot state these data show “rch genetic diversity” First of all WHAT do you mean by “rich” ? Certainly you cannot mean valuable in terms of breeding, second morphological diversity can mean a range of genetic diversity—depending on the genetic control of traits-so you can have no idea how much genetic diversity  exists among these accessions.

Response 7

Agreed. I have made changes based on your comments. (Line 210-229)

Point 8

Effective spikes and thousand kernel weight are important factors in yield composition[45][46]. The morphology and distribution of maize leaves have a direct impact on the interception of light energy by the population canopy and the efficiency of photosynthetic utilization, which ultimately affects yield formation[47]. The study found a significant or highly significant positive correlation between leaf length and width, and chlorophyll content, effective spikes, and thousand kernel weight. Therefore, it is important to consider leaf blade size during the breeding process. The study analysed 192 maize germplasm resources, which exhibit a high level of diversity, reflecting rich genetic variation. Therefore, the traits can be targeted according to the needs of different breeding objectives.

This is all a gross over-simplification of how these traits may or may not be useful –AGAIN you cannot say “rich genetic diversity”  Traits can be targeted for selection BUT THE KEY TRAIT IS YIELD.

Response 8

Agreed. I have made changes to the content of the article.

Round 3

Reviewer 4 Report

Comments and Suggestions for Authors

REVIEW REV 3

Comprehensive evaluation and selection of 192 maize accessions from different sources 2 3 Mengting Hu, Huijuan Tian, Kaizhi Yang, Shuqi Ding, Ying Hao, Ruohang Xu, Fulai Zhang, Hong Liu, Dan Zhang*

1). This investigation into the diversity and 21 genetic principles of phenotypic traits in maize accessions has pinpointed superior germplasm adaptable to Xinjiang's Aral 22 region.

NO These studies are purely phenotypic they in no way pinpoint superior germplasm as you so not include date for most of the agronomically important traits

2). . However, The varied 37

Delete  However. sentence starts—The varied…

3) Analyzing accessions is a critical step for further research aimed at deciphering their genetic 45 backgrounds.

Analyzing accessions for WHAT, using WHAT methods, examining WHAT?

4)Principal component analysis (PCA), cluster analysis, and GGE biplot emerge as pivotal tools in this research. 46

NO studies of phenotypic primarily non agronomically characteristics provides an INITIAL  MOST CERTAINLY NOT PIVOTAL!!!!!!!!!!!!

5) whereas cluster analysis aids 47 in species classification and agricultural zoning[11][12]. Additionally, GGE biplot analysis not only evaluates regional test 48

NO The distinction is NOT methods but characteristics used in an analysis

6) Multivariate statistical analyses, leveraging the depth of multivariate 50 analysis, have been extensively applied to explore genetic patterns, relationships, and interactions among 51 traits[13][14][15][16].

PRECISELY each of these refs speak to agronomically useful characteristics emphasizing point 5).

7) Numerous studies have focused on the morphological identification and evaluation of maize accessions. Li et al. 53 established the core germplasm of local maize varieties in China, selecting based on geographic origin and phenotypic 54 diversity. Their findings highlighted that varieties from the mountainous southwest exhibited significant phenotypic 55 diversity[17][18]. D.J. Ogunniyan et al. assessed 15 superior yellow maize inbreds, noting that except for panicles per plant 56 and yield, environmental factors minimally influenced the other traits[19]. Taba et al. created a core subset of Caribbean 57 maize from the CIMMYT germplasm bank, evaluating its agronomic and morphological traits[20]. Cai Yi-Lin analyzed the 58 phenotypic diversity of 710 maize accessions by geographic origin, revealing notable regional differences in diversity 59 indices and traits, with germplasm from South China, East China, and Southwest China showing higher diversity than other 60 regions[21]. Iqbal J et al. performed principal component and cluster analyses on 34 agronomic traits from 153 materials, 61 resulting in five clusters and seven principal components[22]. Dong Xin et al. conducted an extensive phenotypic 62 assessment of 129 local maize varieties in Chongqing province over two years, organizing them into three main groups[23]. 63 Meng Zuqing et al. evaluated the phenotypic diversity of 179 local maize varieties in Tibet through multi-year trials, laying 64 the groundwork for the preservation and development of these varieties in Tibet[24]. Wang Dawei et al. analyzed mid- to 65 late-maturing maize varieties over two years, identifying several principal components and employing a GGE biplot to 66 analyze yields, demonstrating the benefits and limitations of subgroup analyses for repeated trials[25]. Xie Wenjin et al. 67 classified silage maize in the Huanghuaihai region using principal component analysis, identifying two varieties as suitable 68 for most areas through GGE biplot analysis[26]. Shojaei S H et al. utilized GGE biplot analysis to assess the stability of 12 69 maize hybrids, identifying a material that combines high yield with stability[27].

None of the refs in this chapter that have analysed agronomically useful traits are relevant to be quoted in this paper—those that refer only to phenotypic diversity are.

8) The F-value is indicative of the overall trait qual 162 ity of the accessions, with a higher F-value reflecting a superior comprehensive trait performance of the germplasm pheno 163 type, and the opposite indicating inferior performance. Upon analyzing the phenotypic traits of the maize accessions, the 164 average comprehensive performance was calculated at 0.50. Accession No. 108 exhibited the highest F-value (0.77), show 165 casing the best comprehensive performance, whereas Accession No. 150 displayed the lowest F-value (0.13), denoting the 166 poorest comprehensive performance(Appendix C). T

You have not indicate on what basis you score trait quality. In any event looking only at specific phenotypic traits without linking genetically to agronomic performance attributes CANNOT provide any basis for determining trait quality. Your data do not and cannot support any conclusions about trait quality/ All references to trait quality should be deleted.

9) Cluster analysis facilitates the grouping of accession metrics, enabling the classification of genetically similar accessio 183 ns into homogeneous groups.

NO NO NO as per 8) you cannot use these data to determine genetically similar accessions as you do not know the genetic basis of the phenotypic traits

10) Agronomic trait diversity analysis is a crucial instrument for examining crop genetic diversity[38][38]. Investigating the 324 diversity of agronomic traits in maize accessions can uncover genetic variances among traits from diverse sources, 325 facilitating the identification of germplasm with superior performance.

NO NO NO as previously said i) you have for the most part NOT examined agronomically useful traits, ii) at least this study is one INITIAL means to characterise this germplasm for further studies,  iii) while such studies can or might uncover genetic variance you imply that these studies actually do that BUT THEY DO NOT

11) . Thus, breeding efforts should focus on selecting 346 and enhancing plant traits to effectively increase yield.

If all this study has done is to allow you to state the obvious then what is the point of this publication!

12) The 192 maize germplasm resources analyzed displayed significant 347 phenotypic diversity, indicating that traits can be selectively screened to meet various breeding objectives.T

NO NO NO you have not showed nor do you speak to the relative importance of the specific traits you have analysed in contributing to yield.  

13) Maize accessions can thus be comprehensively ranked based on 361 the evaluation values derived from both the principal component scores and affiliation function values.

NO –HOW???????????? You do not have the data to show how !

14) Group III exhibited 366 superior performance, ranking first among the three groups, making it the most favorable option for cultivation in the Aral 367 region of Xinjiang and areas with similar climates.T

As repeatedly said in these comments you have not shown HOW the combination of traits shown by group III “are most favorable for cultivation in Aral”

TO counter these criticisms, you need to state upfront what the favorable traits for fodder or silage maize are and cite all those refs—THEN you can be in a position to evaluate how specific accessions meet or fail to meet those criteria BUT without any  refence to genetic control of individual traits you CAN NEVER be in a position to state which accessions should be crossed to develop superior germplasm

15) Climatic conditions, which can 403 significantly vary between years, profoundly influence the growth and development of materials. The selection of varieties 404 necessitates adaptability to the ecological environment, stability, and suitability for growth and development in the 405 targeted environment, ensuring the identification of high-quality varieties with greater reliability. Therefore, determining 406 the relative stability of the same material across different years is crucial for screening high-quality and stable materials.

NO I have said before that 2 years of data is totally insufficient to speak to determine stability of traits expression across years.

16) Through screening, 10 stable and high biomass maize germplasms were identified among 192 accessions. Yield 411 traits, a primary focus for breeders, and the stable materials identified for yield-related traits in this study can support the 412 breeding of high-yielding and stable varieties in later stages

YOU MUST qualify what you mean by yield  IF you always mean fodder or silage yield you must state this otherwise if you only speak to yield readers will automatically think you are talking about grain yield.

Author Response

Hello, thank you very much for your guidance on my essay in your busy schedule, for which I feel honoured and happy. At the same time, I apologise for the inefficiency in changing the article, which was only completed near the deadline. My experimental data were 295 maize germplasm with three replications and three plots. The reason why there are only phenotypic traits, which causes this problem, is because we intend to explore these germplasm initially, and you are right, it is true that only the most primitive and basic research has been carried out. Because these germplasm come from various countries with different adaptations to the environment, and there is a lot of variation between the germplasm, it is true that such a large amount of work on agronomic traits is a bit overwhelming, but equally, this has resulted in a lot of germplasm where most of the good traits have not been discovered or studied. Another reason I am sorry is that the last time I did a touch-up on the English quality, many of the statements made were ambiguous, which led to a deviation in your understanding, and I apologise for that. Also, I wish you good health and a happy life. Looking forward to your next comment.

Point 1

This investigation into the diversity and 21 genetic principles of phenotypic traits in maize accessions has pinpointed superior germplasm adaptable to Xinjiang's Aral 22 region.

NO These studies are purely phenotypic they in no way pinpoint superior germplasm as you so not include date for most of the agronomically important traits

Response 1

Agreed. Thank you for your comment, it has been modified. Line 22-24

Point 2

However, The varied 37

Delete  However. sentence starts—The varied…

Response 2

Agreed. I have deleted it. Line 32-33.

Point 3

Analyzing accessions is a critical step for further research aimed at deciphering their genetic 45 backgrounds.

Analyzing accessions for WHAT, using WHAT methods, examining WHAT?

Response 3

Agreed. I have amended it. I am very sorry, this part is probably due to an ambiguity that occurred while making English quality polish. This resulted in an error in the statement.

Point 4

Principal component analysis (PCA), cluster analysis, and GGE biplot emerge as pivotal tools in this research. 46

NO studies of phenotypic primarily non agronomically characteristics provides an INITIAL  MOST CERTAINLY NOT PIVOTAL!!!!!!!!!!!!

Response 4

Agreed. I have amended it. Related content about GGE biplot has been removed。

Point 5

whereas cluster analysis aids 47 in species classification and agricultural zoning[11][12]. Additionally, GGE biplot analysis not only evaluates regional test 48

NO The distinction is NOT methods but characteristics used in an analysis

Response 5

Agreed. I have made revisions. I have made additions and corrections to the preamble.

Point 6

Multivariate statistical analyses, leveraging the depth of multivariate 50 analysis, have been extensively applied to explore genetic patterns, relationships, and interactions among 51 traits[13][14][15][16].

PRECISELY each of these refs speak to agronomically useful characteristics emphasizing point

Response 6

Agreed. I have made revisions.

Point 7

Numerous studies have focused on the morphological identification and evaluation of maize accessions. Li et al. 53 established the core germplasm of local maize varieties in China, selecting based on geographic origin and phenotypic 54 diversity. Their findings highlighted that varieties from the mountainous southwest exhibited significant phenotypic 55 diversity[17][18]. D.J. Ogunniyan et al. assessed 15 superior yellow maize inbreds, noting that except for panicles per plant 56 and yield, environmental factors minimally influenced the other traits[19]. Taba et al. created a core subset of Caribbean 57 maize from the CIMMYT germplasm bank, evaluating its agronomic and morphological traits[20]. Cai Yi-Lin analyzed the 58 phenotypic diversity of 710 maize accessions by geographic origin, revealing notable regional differences in diversity 59 indices and traits, with germplasm from South China, East China, and Southwest China showing higher diversity than other 60 regions[21]. Iqbal J et al. performed principal component and cluster analyses on 34 agronomic traits from 153 materials, 61 resulting in five clusters and seven principal components[22]. Dong Xin et al. conducted an extensive phenotypic 62 assessment of 129 local maize varieties in Chongqing province over two years, organizing them into three main groups[23]. 63 Meng Zuqing et al. evaluated the phenotypic diversity of 179 local maize varieties in Tibet through multi-year trials, laying 64 the groundwork for the preservation and development of these varieties in Tibet[24]. Wang Dawei et al. analyzed mid- to 65 late-maturing maize varieties over two years, identifying several principal components and employing a GGE biplot to 66 analyze yields, demonstrating the benefits and limitations of subgroup analyses for repeated trials[25]. Xie Wenjin et al. 67 classified silage maize in the Huanghuaihai region using principal component analysis, identifying two varieties as suitable 68 for most areas through GGE biplot analysis[26]. Shojaei S H et al. utilized GGE biplot analysis to assess the stability of 12 69 maize hybrids, identifying a material that combines high yield with stability[27].

None of the refs in this chapter that have analysed agronomically useful traits are relevant to be quoted in this paper—those that refer only to phenotypic diversity are.

Response 7

Agreed. I have revised. Agronomic traits are the ultimate expression of phenotypic traits and many papers combine them. This part of the introduction I tend to be more about the differences in research methods, and the analyses used in my thesis are presented in these literatures.

Point 8

The F-value is indicative of the overall trait qual 162 ity of the accessions, with a higher F-value reflecting a superior comprehensive trait performance of the germplasm pheno 163 type, and the opposite indicating inferior performance. Upon analyzing the phenotypic traits of the maize accessions, the 164 average comprehensive performance was calculated at 0.50. Accession No. 108 exhibited the highest F-value (0.77), show 165 casing the best comprehensive performance, whereas Accession No. 150 displayed the lowest F-value (0.13), denoting the 166 poorest comprehensive performance(Appendix C). T

You have not indicate on what basis you score trait quality. In any event looking only at specific phenotypic traits without linking genetically to agronomic performance attributes CANNOT provide any basis for determining trait quality. Your data do not and cannot support any conclusions about trait quality/ All references to trait quality should be deleted.

Response 8

Agreed. This section has been deleted. But I think that the comprehensive evaluation F-value can be used to evaluate germplasm and sort it. This is because there is a lot of literature on combining principal components with the affiliation function for comprehensive evaluation. Although my data did not include most of the agronomic traits, it is just a method with the help of which the phenotypes of maize germplasm can be evaluated and ranked, and there is consistency between the results of the affiliation function and the results of the cluster analysis in most of the studies. My analyses were no exception.

Point 9

Cluster analysis facilitates the grouping of accession metrics, enabling the classification of genetically similar accessio 183 ns into homogeneous groups.

NO NO NO as per 8) you cannot use these data to determine genetically similar accessions as you do not know the genetic basis of the phenotypic traits

Response 9

Agreed. It has been modified. This issue was mentioned by you last time. I'm very sorry that it came up again this time, probably because there was a deviation in the translation. It caused ambiguity.

Point 10

Agronomic trait diversity analysis is a crucial instrument for examining crop genetic diversity[38][38]. Investigating the 324 diversity of agronomic traits in maize accessions can uncover genetic variances among traits from diverse sources, 325 facilitating the identification of germplasm with superior performance.

NO NO NO as previously said i) you have for the most part NOT examined agronomically useful traits, ii) at least this study is one INITIAL means to characterise this germplasm for further studies,  iii) while such studies can or might uncover genetic variance you imply that these studies actually do that BUT THEY DO NOT

Response 10

Agreed. I've revised it.

Point 11

Thus, breeding efforts should focus on selecting 346 and enhancing plant traits to effectively increase yield.

If all this study has done is to allow you to state the obvious then what is the point of this publication!

Response 11

Agreed. I've revised it.

Point 12

The 192 maize germplasm resources analyzed displayed significant 347 phenotypic diversity, indicating that traits can be selectively screened to meet various breeding objectives.T

NO NO NO you have not showed nor do you speak to the relative importance of the specific traits you have analysed in contributing to yield.

Response 12

Agreed. I've revised it.

Point 13

Maize accessions can thus be comprehensively ranked based on 361 the evaluation values derived from both the principal component scores and affiliation function values.

NO –HOW???????????? You do not have the data to show how !

Response 13

Agreed.

Point 14

 Group III exhibited 366 superior performance, ranking first among the three groups, making it the most favorable option for cultivation in the Aral 367 region of Xinjiang and areas with similar climates.T

As repeatedly said in these comments you have not shown HOW the combination of traits shown by group III “are most favorable for cultivation in Aral”

TO counter these criticisms, you need to state upfront what the favorable traits for fodder or silage maize are and cite all those refs—THEN you can be in a position to evaluate how specific accessions meet or fail to meet those criteria BUT without any  refence to genetic control of individual traits you CAN NEVER be in a position to state which accessions should be crossed to develop superior germplasm

Response 14

The means and coefficients of variation for each group are presented in Table 4, where I mentioned the characteristics of the group. Because he performs well in this environment and is well adapted.

Point 15

Climatic conditions, which can 403 significantly vary between years, profoundly influence the growth and development of materials. The selection of varieties 404 necessitates adaptability to the ecological environment, stability, and suitability for growth and development in the 405 targeted environment, ensuring the identification of high-quality varieties with greater reliability. Therefore, determining 406 the relative stability of the same material across different years is crucial for screening high-quality and stable materials.

NO I have said before that 2 years of data is totally insufficient to speak to determine stability of traits expression across years.

Response 15

Agreed. I've revised it.

Point 16

Through screening, 10 stable and high biomass maize germplasms were identified among 192 accessions. Yield 411 traits, a primary focus for breeders, and the stable materials identified for yield-related traits in this study can support the 412 breeding of high-yielding and stable varieties in later stages

YOU MUST qualify what you mean by yield  IF you always mean fodder or silage yield you must state this otherwise if you only speak to yield readers will automatically think you are talking about grain yield.

Response 16

Agreed. As the stability section is deleted. So that section has been deleted. You have a point, and indeed it should be stated exactly which part of the yield is being produced, otherwise it can be ambiguous.

Round 4

Reviewer 4 Report

Comments and Suggestions for Authors

Your sentences following need to be revised particularly because "intuitiveness" in this context is NOT correct  and see other edit{Phenotypic traits, re- 42 flecting gene-environment interactions, are advantageous due to their intuitiveness and 43 convenience[9]. Enhancements in plant structure can notably augment maize kernel yield. 44 As noted by Malik et al. and Rahman et al., critical agronomic traits for maize encompass 45 plant height, ear height, stem thickness, leaf width, leaf length, total leaf count, ear count 47 per plant, thousand kernel weight, and male ear branching[10][11].

REVISE TO

. Phenotypic traits, re- 42 flecting gene-environment interactions, are readily available and so convenient for evaluation [9]. Enhancements in plant structure can notably augment maize kernel yield and, as noted by Malik et al. and Rahman et al., critical agronomic traits for maize encompass 45 .......as in current text

Comments on the Quality of English Language

.Your sentences following need to be revised particularly because "intuitiveness" in this context is NOT correct  and see other edit{Phenotypic traits, re- 42 flecting gene-environment interactions, are advantageous due to their intuitiveness and 43 convenience[9]. Enhancements in plant structure can notably augment maize kernel yield. 44 As noted by Malik et al. and Rahman et al., critical agronomic traits for maize encompass 45 plant height, ear height, stem thickness, leaf width, leaf length, total leaf count, ear count 47 per plant, thousand kernel weight, and male ear branching[10][11].

REVISE TO

. Phenotypic traits, re- 42 flecting gene-environment interactions, are readily available and so convenient for evaluation [9]. Enhancements in plant structure can notably augment maize kernel yield and, as noted by Malik et al. and Rahman et al., critical agronomic traits for maize encompass 45 ......as in current text

Author Response

Thanks very much for your advice.

Point 1

Your sentences following need to be revised particularly because "intuitiveness" in this context is NOT correct  and see other edit{Phenotypic traits, re- 42 flecting gene-environment interactions, are advantageous due to their intuitiveness and 43 convenience[9]. Enhancements in plant structure can notably augment maize kernel yield. 44 As noted by Malik et al. and Rahman et al., critical agronomic traits for maize encompass 45 plant height, ear height, stem thickness, leaf width, leaf length, total leaf count, ear count 47 per plant, thousand kernel weight, and male ear branching[10][11].

REVISE TO

.Phenotypic traits, re- 42 flecting gene-environment interactions, are readily available and so convenient for evaluation [9]. Enhancements in plant structure can notably augment maize kernel yield and, as noted by Malik et al. and Rahman et al., critical agronomic traits for maize encompass 45 .......as in current text。

Response 1

Agreed. I have modified. line 42
